# ON ADVERSARIAL TRAINING WITHOUT PERTURBING ALL EXAMPLES

**Max Losch**[1]**, Mohamed Omran**[1]**, David Stutz**[1]**, Mario Fritz**[2]**, Bernt Schiele**[1]
[1] Max Planck Institute for Informatics, Saarland Informatics Campus,
[2] CISPA Helmholtz Center for Information Security, Saarbrücken
`{mlosch, mohomran, dstutz, schiele}@mpi-inf.mpg.de, fritz@cispa.de`

## ABSTRACT

Adversarial Training (AT) is the de-facto standard for improving robustness against adversarial examples. This usually involves a multi-step adversarial attack applied on each example during training. In this paper, we explore only constructing Adversarial Examples (AEs) on a subset of the training examples. That is, we split the training set in two subsets $A$ and $B$, train models on both ($A \cup B$) but construct AEs only for examples in $A$. Starting with $A$ containing only a single class, we systematically increase the size of $A$ and consider splitting by class and by examples. We observe that: (i) adv. robustness transfers by difficulty and to classes in $B$ that have never been adv. attacked during training, (ii) we observe a tendency for hard examples to provide better robustness transfer than easy examples, yet find this tendency to diminish with increasing complexity of datasets (iii) generating AEs on only $50\%$ of training data is sufficient to recover most of the baseline AT performance even on ImageNet. We observe similar transfer properties across tasks, where generating AEs on only $30\%$ of data can recover baseline robustness on the target task. We evaluate our subset analysis on a wide variety of image datasets like CIFAR-10, CIFAR-100, ImageNet-200 and show transfer to SVHN, Oxford-Flowers-102 and Caltech-256. In contrast to conventional practice, our experiments indicate that the utility of computing AEs varies by class and examples and that weighting examples from $A$ higher than $B$ provides high transfer performance. Code is available at `http://github.com/mlosch/SAT`.

## 1 INTRODUCTION

Imperceptible changes to an image can change the output of a well-performing classification model dramatically. These so-called Adversarial Examples (AEs) have been the focus of a large body on deep learning vulnerabilities of works since its discovery (Szegedy et al., 2014). To date, Adversarial Training (AT) (Madry et al., 2018; Carlini et al., 2019) and its variants (Zhang et al., 2019; Carmon et al., 2019; Wu et al., 2020) is the de-facto state-of-the-art in improving the robustness against AEs. Essentially, AT generates adversarial perturbations for all examples seen during training. While adversarial training is known to transfer robustness to downstream tasks (Shafahi et al., 2020; Salman et al., 2020; Yamada & Otani, 2022) and that robustness is distributed unevenly across classes (Tian et al., 2021; Xia et al., 2021), common practice dictates that AT "sees" adversarial examples corresponding to the whole training data, including all classes and concepts therein. This is independent of whether only adversarial robustness is optimized or a trade-off between robustness and clean performance is desired (Stutz et al., 2019). This also holds for variants that treat individual examples differently (Ding et al., 2020; Wang et al., 2020; Kim et al., 2021) or adaptively select subsets to attack during training to reduce computational overhead (Hua et al., 2021; Dolatabadi et al., 2022). In this common setup, it is thus difficult to ascertain if a generic, transferable robustness is learned by the model, or if adversarial robustness is learned separately for each class in the training set. In the worst case, a model that can only learn to be robust on a class-specific basis would have to be presented with adversarially-perturbed images for every class of interest. A more preferable outcome, on the other hand, would be a model with *systematic adversarial robustness* (Bahdanau et al., 2019; Omran & Schiele, 2022): i.e. a model that systematically generalizes to novel combinations of perturbations and classes that were separately encountered during training.

Figure 1: Adversarial robustness transfers among classes. Using Subset Adversarial Training (SAT), during which only a subset of all training examples ($A$) are attacked, we show that robust training even on a single class provides robustness transfer to all other, non adv. trained, classes ($B$). E.g., SAT for $A$=cat, we observe an robust accuracy of 37.8% on $B$. Noteworthy is the difference of transfer utility between classes. I.e. $A$=car provides very little transfer to $B$ (17.1%). We investigate this transfer among classes and provide new insights for robustness transfer to downstream tasks.

To shed light on this issue, we consider an analysis setup depicted in figure 1, we coin Subset Adversarial Training (SAT), where we split the training data into two subsets $A$ and $B$, train the model conventionally on the union ($A \cup B$), but generate AEs only on examples from $A$ (indicated by the emoji). For example, we can split training data by class, with $A = \{car\}$ or $A = \{cat\}$ and $B = A^c$, and investigate how adversarial robustness transfers. Surprisingly, we observe significant adversarial robustness on $B_{\text{val}}$ at test time, the degree of which depends on the class(es) in $A$. Of course, $A$ and $B$ can be arbitrary partitions of the training data. For example, we could put only "difficult" examples in $A$ during training. At test time, we evaluate overall adversarial robustness (since there is no natural split into $A_{\text{val}}$ or $B_{\text{val}}$). These experiments reveal a rather complex interaction of adversarial robustness between classes and examples.

Our analysis provides a set of **contributions** revealing a surprising generalizability of robustness towards non-adv. trained classes and examples even under scarce training data setups. **First**, selecting subsets of whole classes, we find that SAT provides transfer of adversarial robustness to classes which have never been attacked during training. E.g. only generating adversarial examples for class *car* on CIFAR-10, achieves a non-trivial robust accuracy of 17.1% on all remaining CIFAR-10 classes (see figure 1, right). **Secondly**, we observe classes and examples that are hard to classify do generally provide better robustness transfer than easier ones. I.e. class *cat* achieves more than twice the robust accuracy on the remaining classes (37.8%) over class *car* (17.1%). **Thirdly**, SAT with 50% of training data is sufficient to recover the baseline performance with vanilla AT even on hard datasets like ImageNet. **Fourthly**, we observe similar transfer properties of SATed models to downstream tasks. In this setting, exposing the model to only 30% of AEs during training, can recover baseline AT performance on the target task. **Lastly**, while we observe promising amounts of transfer, there is room for improvement. Our SAT approach can be viewed as an analytical tool that can foster the development of models that exhibit stronger systematic generalization to adversarial perturbations.

## 2 RELATED WORK

Since their discovery (Szegedy et al., 2014), robustness against adversarial examples has mainly been tackled using adversarial training (Goodfellow et al., 2015; Madry et al., 2018; Zhang et al., 2019). Among many others, prior work proposed adversarial training variants working with example-dependent threat models (Balaji et al., 2019; Ding et al., 2020; Wang et al., 2020; Kim et al., 2021), acknowledging that examples can have different difficulties. Some works also mine hard examples (Hua et al., 2021) or progressively prune a portion of the training examples throughout training (Dolatabadi et al., 2022; Kaufmann et al., 2022). However, all of these methods generally assume access to adversarial examples on the whole training set. That is, while individual examples can be dropped during training or are treated depending on difficulty, the model can see adversarial perturbations for these examples if deemed necessary. Adversarial training is also known to transfer robustness to downstream tasks (Salman et al., 2020; Yamada & Otani, 2022; Shafahi et al., 2020) and adversarially robust representations can be learned in a self-supervised fashion (Gowal et al., 2021). Here, a robust backbone is often adapted to the target task by re-training a shallow classifier – sometimes in an adversarial fashion. It is generally not studied whether seeing adversarial examples on the whole training set is required for good transfer. This is despite evidence that achieving adversarial robustness is easier for some classes/concepts than for others (Benz et al., 2020; Nanda et al., 2021; Xia et al., 2021; Tian et al., 2021), also for robustness transfer (Jain et al., 2023). Complementing these works, we consider only constructing adversarial examples on a pre-defined subset of the training set, not informed by the model or training procedure, and study how robustness transfers across examples and tasks. Our work is thus related to the broader problem of measuring

systematic generalization and considering the effects of spurious correlations (Bahdanau et al., 2019; Ruis et al., 2020; Geirhos et al., 2020; Montero et al., 2021; Schott et al., 2022; Omran & Schiele, 2022).

## 3 BACKGROUND AND METHOD

### 3.1 ADVERSARIAL TRAINING (AT)

It is a well known fact that conventional deep networks are vulnerable to small, often imperceptible, changes in the input. As mitigation, AT is a common approach to extend the empirical risk minimization framework (Madry et al., 2018). Let $(x, y) \in \mathcal{D}_{\text{train}}$ be a training set of example and label pairs and $\theta$ be trainable parameters, then AT is defined as:

$$\min_{\theta} \mathbb{E}_{(x,y) \in \mathcal{D}_{\text{train}}} \left[ \max_{||\delta||_p \leq \epsilon} \mathcal{L}(x + \delta, y; \theta) \right], \tag{1}$$

where $\delta$ is a perturbation that maximizes the training loss $\mathcal{L}$ and thus training error. The idea being that, simultaneously to minimizing the training loss, the loss is also optimized to be stable within a small space $\epsilon$ around each training example $||\delta||_p \leq \epsilon$, $p \geq 1$. We consider the $L_2$ norm in our main paper and provide results for $L_\infty$ in the appendix A.8. This additional inner maximization is solved by an iterative loop; conventionally consisting of 7 or more steps. In some settings (Goodfellow et al., 2015; Stutz et al., 2019; Zhang et al., 2019), the robust loss is combined with the corresponding loss on clean examples in a weighted fashion to control the trade-off between adversarial robustness and clean performance.

### 3.2 AT WITHOUT PERTURBING ALL TRAINING EXAMPLES

Most proposed AT methodologies generate AEs on the whole training set. This being also valid for methods which adaptively select subsets (Hua et al., 2021; Dolatabadi et al., 2022) during training, for adaptive attack iterations (Zhang et al., 2020), for adaptive example weightings (Huang et al., 2020; Zhang et al., 2021; Ge et al., 2023) or more traditional AT in which only a subset per batch is adversarially attacked. These methods do not guarantee the exclusion of examples, that is, the model is likely to see an AE for every example in the training set. From a broader perspective, the necessity to generate AEs exhaustively for all classes appears unfortunate though. Ideally, we desire robust models to be scalable, i.e. transfer flexibly from few examples and across classes to unseen ones (Omran & Schiele, 2022). We propose SAT to investigate to what extent AT provides this utility. To formalize, let $A$ be a training subset and $B$ contain the complement: $A \subset \mathcal{D}_{\text{train}}, B = \mathcal{D}_{\text{train}} \setminus A$. Then SAT applies the inner maximization loop of AT on the subset $A$ only; on $B$ the conventional empirical risk is minimized:

$$\min_{\theta} \mathbb{E}_{(x,y) \in \mathcal{D}_{\text{train}}} \left[ w_A \mathbb{1}_{(x,y) \in A} \max_{||\delta||_2 \leq \epsilon} \mathcal{L}(x + \delta, y; \theta) + w_B \mathbb{1}_{(x,y) \in B} \mathcal{L}(x, y; \theta) \right], \tag{2}$$

where $\mathbb{1}_{(x,y) \in A}$ is 1 when the training example is in $A$ and 0 otherwise. $w_A$ and $w_B$ define optional weights, which are by default both set to 1. Note that this is different from balancing robust and clean loss as discussed in (Goodfellow et al., 2015; Stutz et al., 2019; Zhang et al., 2019), where the model still encounters adversarial examples on the whole training set.

**Loss balancing.** The formulation in equation 2 implies an imbalance between left and right loss when the training split is uneven ($|A| \neq |B|$). To counteract, we assign different values to $w_A$ and $w_B$ based on their subset size. E.g., to equalize, we assign $w_B = 1$ and $w_A = |B|/|A|$. We will utilize this loss balancing to improve robustness for transfer learning in section 4.3.

### 3.3 TRAINING AND EVALUATION RECIPES

Consider the depiction of SAT in figure 1. Prior to training, the training set is split into $A$ and $B$ (left). For evaluation (middle), we split the validation set into a corresponding split of $A_{\text{val}}$ and $B_{\text{val}}$, if possible. For **Class-subset Adversarial Training (CSAT)**, this split aligns with the classes on the dataset: $A$ and $B$ are all training examples corresponding to two disjoint sets of classes while $A_{\text{val}}$

and $B_{\text{val}}$ are the corresponding test examples of these classes. As experimenting with all possible splits of classes is infeasible, we motivate splits by class difficulty where we measure difficulty by the average entropy of predictions per class – introduced as $\mathcal{H}_C$ in the next paragraph. In contrast, we can also split based on individual example difficulty. We provide empirical support for this approach in the experimental section 4. Additionally, example difficulty has been frequently linked to proximity between decision boundary and example (Baldock et al., 2021; Ding et al., 2020; Kim et al., 2021; Hua et al., 2021; Agarwal et al., 2022). The closer the example is to the boundary, the harder it is likely to classify. The hypothesis: hard examples provide a larger contribution to training robust models, since they optimize for large margins (Ding et al., 2020; Wang et al., 2020). We refer to this experiment as **Example-subset Adversarial Training (ESAT)**. In contrast to CSAT, however, there is no natural split of the test examples into $A_{\text{val}}$ and $B_{\text{val}}$ such that we evaluate robustness on the whole test set (i.e., $\mathcal{D}_{\text{val}}$).

As difficulty metric, we utilize entropy over softmax, which we empirically find to be as suitable as alternative metrics (discussed in the supplement, section A.2). Consider a training set example $x \in \mathcal{D}_{train}$ and a classifier $f$ mapping from input space to logit space with $N$ logits. Then the entropy of example $x$ is determined by $\mathcal{H}(f(x))$ and of a whole class $C \subset \mathcal{D}_{train}$ is determined by $\mathcal{H}_C(f)$, the average over all examples in $C$:

$$\mathcal{H}(f(x)) = -\sum_{i=1}^{N} \sigma_i(f(x)) \cdot \log \sigma_i(f(x)), \quad \mathcal{H}_C(f) = \frac{1}{|C|} \sum_{x \in C} \mathcal{H}(f(x)),$$

where $\sigma$ denotes the softmax function. For our SAT setting, we rank examples prior to adversarial training. This requires a classifier pretrained on $\mathcal{D}_{train}$ enabling the calculation of the entropy. To strictly separate the effects between entropy and AT, we determine the entropy using a non-robust classifier trained without AT. Similar to (Agarwal et al., 2022), we aggregate the classifier states at multiple epochs during training and average the entropies. Let $f_1, f_2, ... f_M$ be snapshots of the classifier from multiple epochs during training, where $M$ denotes the number of training epochs. Then the average entropy for an example is given by $\overline{\mathcal{H}}(x)$ and for a class by $\overline{\mathcal{H}}_C(f)$:

$$\overline{\mathcal{H}}(x) = \frac{1}{M} \sum_{e=1}^{M} \mathcal{H}(f_e(x)), \quad \overline{\mathcal{H}}_C = \frac{1}{M} \sum_{e=1}^{M} \mathcal{H}_C(f_e). \tag{3}$$

## 4 EXPERIMENTS

As aforementioned, common practice performs AT for the whole training set. In the following, we explore CSAT and ESAT, which splits the training set in two subsets $A$ and $B$ and only constructs AEs for $A$ such that the model never sees AEs for $B$. We start with single-class CSAT – $A$ contains only examples of a single class – and increase the size of $A$ (section 4.1) by utilizing the entropy ranking of classes $\mathcal{H}_C$ (equation 3). ESAT, which splits into example subsets is discussed in section 4.2. Both SAT variants reveal complex interactions between classes and examples while indicating that few AEs provide high transfer performance to downstream tasks when weighted appropriately (section 4.3).

**Training and evaluation details.** Since AT is prone to overfitting (Rice et al., 2020), it is common practice to stop training when robust accuracy on a hold-out set is at its peak. This typically happens after a learning rate decay. We adopt this "early stopping" for all our experiments by following the methodology in (Rice et al., 2020) but utilize Auto Attack (AA) to evaluate robust accuracy. Throughout the course of the training, we evaluate AA after each learning rate decay on $10\%$ of validation data $\mathcal{D}_{\text{val}}$ and perform a final evaluation with the model providing the highest robust accuracy. This evaluation is performed on the remaining $90\%$ of validation data. This AA split is fixed throughout experiments to provide consistency. If not specified otherwise, we generate adversarial examples during training with PGD-7 within an $L_2$ epsilon ball of $\epsilon = 0.5$ (all CIFAR variants) or $\epsilon = 3.0$ (all ImageNet variants) – typical configurations found in related work. We train all models from scratch and use ResNet-18 (He et al., 2016) for all CIFAR-10 and CIFAR-100 (Krizhevsky et al., 2009) experiments and ResNet-50 for all ImageNet-200 experiments. Here, ImageNet-200 corresponds to the ImageNet-A subset (Hendrycks et al., 2021) to render random baseline experiments tractable (to reduce training time). This ImageNet-200 dataset, contains 200 classes that retain the class variety and breadth of regular ImageNet, but remove classes that are similar to each other (e.g. fine-grained dog types). We use all training and validation examples from

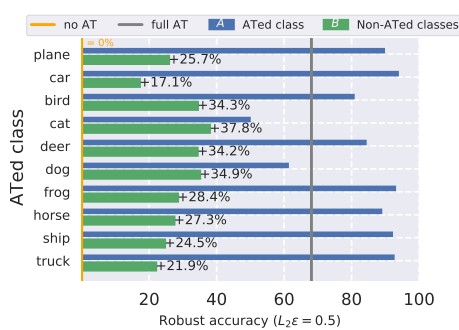 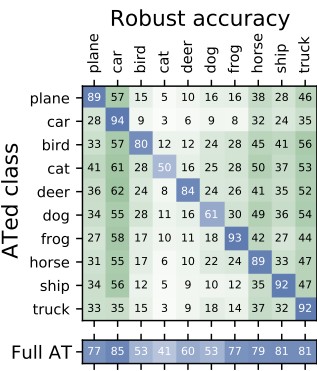

Figure 2: CSAT on a single CIFAR-10 class $A$ (blue), we observe non-trivial transfer to the non-adv. trained classes $B$ (green). Classes considered hard in CIFAR-10 (cat) offer best generalization ($+37.8\%$ gain on non-adv. trained), while easy classes offer the worst (car, $+17.1\%$ gained). Note that without AT, robust accuracy is close to $0\%$ for all classes (orange). Right: same as left, but robust accuracy is evaluated per class (along columns). Here, we observe an unexpected transfer property: hard classes provide better transfer to seemingly unrelated classes (cat $\rightarrow$ truck: $53\%$) than related classes (car $\rightarrow$ truck: $35\%$). Additional results for $\epsilon = 0.25$ and $\epsilon = 1.0$ in the appendix, section A.3.

ImageNet (Deng et al., 2009) that correspond to this subset classes. All training details can be found in the supplement, section A.1.

## 4.1 CLASS SUBSET SPLITS

We start by investigating the interactions between individual classes in $A$ using CSAT on CIFAR-10, followed by an investigation on increasing the number of classes. **Single-class subsets (CSAT).** We train all possible, single class CSAT runs (10) and evaluate robust accuracies on the adv. trained class (A) and the non-adv. trained classes (B). The results are shown in figure 2, left. Each row represents a different training run. Note that the baseline robust accuracy, trained without AT achieves practically $0\%$ (indicated by orange line). Most importantly, we observe non-trivial robustness gains for all classes that have never been attacked during training ($B$-sets). That is, irrespective of the chosen class, we gain at least $17.1\%$ robust accuracy (A=car) on the remaining classes and can gain up to $37.8\%$ robust accuracy when A=cat. These robustness gains are unexpectedly good, given many features of the non-adv. trained classes can be assumed to not be trained robustly. This is consistent across different values for $\epsilon$, as we show in the appendix section A.3.

To investigate this phenomenon further, we analyze robust gains for each individual class and present robust accuracies in the matrix in figure 2, right, where training runs are listed in rows and robust accuracies per class are listed in columns. Blue cells denote the adv. trained class and green cells denote non-adv. trained classes. While we see some expected transfer properties, e.g. CSAT on *car* provides greater robust accuracy on the related class *truck* ($35\%$) than unrelated animal classes *bird, cat, deer, dog* (between $5\%$ and $16\%$), the reverse is not straight-forward. CSAT on *bird* provides $56\%$ robust accuracy on the seemingly unrelated class *truck*, $20\%$-points more than CSAT on *car*. More generally, animal classes provide stronger robustness throughout all classes than inanimate classes. We observe, that these classes are also harder to classify and have a higher entropy $\overline{\mathcal{H}}_C$ as

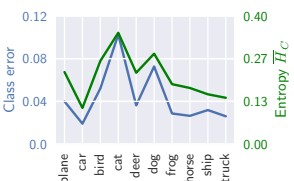

Figure 3: The hardest classes (blue) have the highest entropy (green).

shown in figure 3. This influence of class-entropy might also be utilized to augment the dataset, by adding high entropy samples. We provide a simple proof-of-concept in the appendix, section A.4, adding an 11th class to CIFAR-10.

**Many-class subsets (CSAT).** To increase the number of classes in $A$ while maintaining a minimal computational complexity, we utilize the average class entropy $\overline{\mathcal{H}}_C$ proposed in equation 3 to inform us which ranking to select from. To improve clarity, we begin with a reduced set of experiments on CIFAR-10 before transitioning to larger datasets. We utilize the observed correlation between

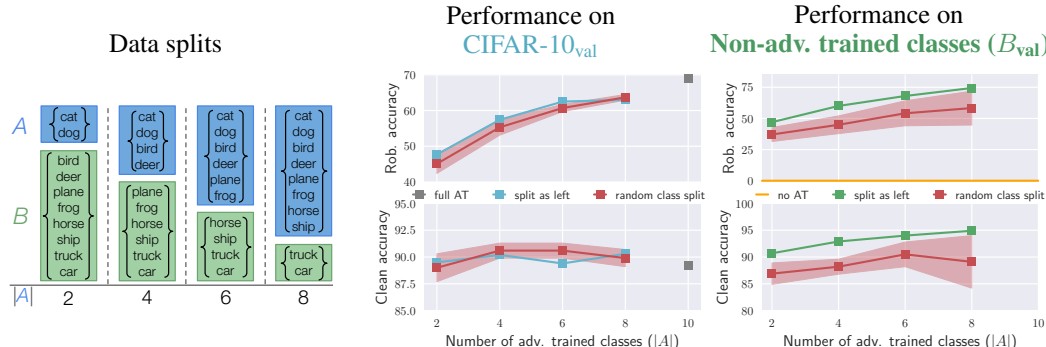

Figure 4: Ranking CIFAR10 classes by difficulty (using entropy as proxy), we perform CSAT with an increasing size of adv. trained classes in $A$. Class splits used for training ($A$ and $B$) are stated on the left. The resulting robust and clean accuracies on the validation set is shown on the right, separated into performance on $B_{val}$ and *all*. Compared with a random baseline of random class ranking (red), we find the ranking by difficulty to have consistently better transfer to non-adv. trained classes ($B$). Overall, this results in an improved robust accuracy on average over all classes.

class difficulty, average class entropy and robustness transfer $\overline{H}_C$ to rank classes and construct 4 adv. trained subsets. Ranked by class entropy $\overline{\mathcal{H}}_C$, we select 4 subsets showing in figure 4, left. As observed before, *cat* and *dog* are hardest and thus first chosen to be in subset $A$. *Truck* and *car* on the other hand are easiest and thus last. To gauge the utility of this ranking, we provide a robust and clean accuracy comparison with a random baseline in figure 4, center and right. I.e., for each subset $A$ we select 10 random subsets and report mean and std. deviation (red line and shaded area). Similar to the single-class setup, we observe subsets of the hardest classes to consistently outperform the random baseline (upper middle plot), up until a subset size of $|A| = 8$, when it draws even. Also note that the robust accuracy on $B_{val}$ is improved across all splits, thus providing support that harder classes – as initially observed on animate vs inanimate classes – offer greater robustness transfer.

For our experiments on larger datasets like CIFAR-100 and ImageNet-200, we additionally evaluate a third ranking strategy. Beside selecting at random and selecting the hardest first, we additionally compare with selecting the easiest (inverting the entropy ranking). We construct 9 subsets per type of ranking (instead of 4) and report robust accuracies for selecting the easiest classes as well. Results are presented in three columns in figure 5; one dataset per column. As before, we show robust accuracies on the tested dataset (upper row) and robust accuracies on $B_{val}$ (lower row). For CIFAR-10, we calculate mean and std. dev. over 10 runs, for CIFAR-100 over 5 runs and for ImageNet-200 over 3 runs. Selecting hardest first (highest entropy) is marked as a solid line and easiest first (lowest entropy) as a dashed line. First and foremost, we observe that irrespective of the dataset and the size of $A$, we see robustness transfer to $B_{val}$. This transfer remains greatest with classes we consider hard, while easy classes provide the least. Nonetheless, we see diminishing returns of such an informed ranking when dataset complexity is increased. E.g. the gap between dashed and solid line on ImageNet-200 is small and random class selection is on-par with the best. The results are similar on CIFAR-100, as shown in figure 5, middle). Based on these results, entropy ranking and selecting classes provides only slight improvements in general. Importantly though, we continue to see the tendency of increased robustness transfer to $B_{val}$, which we will come back to in section 4.3.

## 4.2 EXAMPLE SUBSET SPLITS (ESAT)

Considering that splits along classes are inefficient in terms of reaching the full potential of adversarial robustness, we investigate ranking examples across the whole dataset (ESAT). We follow with the same setup as before but rank examples – and not classes – by entropy $\overline{\mathcal{H}}$. Since it is not feasible to construct corresponding rankings on the validation set, we cannot gauge robustness transfer to $B_{val}$. Instead, we will test transfer performance to downstream tasks in section 4.3. We consequently report robust accuracy and clean accuracy on the whole validation set in figure 6. Similar results for $L_\infty$ are provided in the appendix, section A.8.

Firstly, note that the increase in robust accuracy is more rapid than with CSAT w.r.t. the size of $A$. AT only on $50\%$ of training data ($25k$ examples on CIFAR and $112k$ on ImageNet-200) and the resulting average robust accuracy is very close to the baseline AT performance (gray line). Secondly, note that the gap between hard (solid line) and easy example selection (dashed line) has substantially

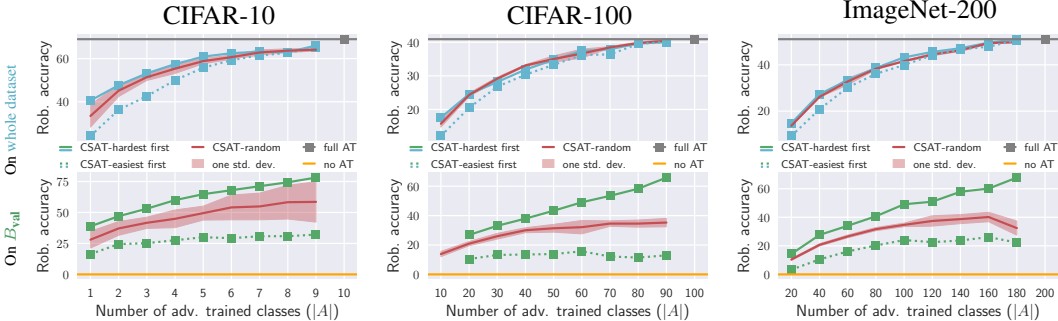

Figure 5: Class-subset Adversarial Training (CSAT) produces non-trivial robustness on classes that have never been attacked during training ($B_{val}$). Along the x-axes we increase the class subset size of $A$ on which AEs are constructed and compare three different class-selection strategies: select hardest first (solid lines), select easiest first (dashed line) and select at random (red). On average, random selection performs as well as informed ranking (upper row), while the robustness transfer to $B_{val}$ is best for the hardest classes (lower row). AT on a single class provides already much greater robust accuracies than without AT (orange).

widened. In practice, it is therefore possible to accidentally select poor performing subsets, although the chance appears to be low given the narrow variance of random rankings (red). To some extent, this observation supports the hypothesis that examples far from the decision border (the easiest to classify) provide the least contribution to robustness gains. This is also supported by the reverse gap in clean accuracy (bottom row in figure 6). That is, easiest-first-selection results in higher clean accuracies than hardest-first, while robust accuracies are much lower. In contrast, however, we observe random rankings (red) to achieve similar performances to hard rankings (solid lines) on all datasets and subset sizes. This is somewhat unexpected, especially on small sizes of $A$ (e.g. $5k$). Given the results, we conjecture that the proximity to the decision boundary plays a subordinate role to increasing robustness. Instead, it is plausible to assume that diversity in the training data has a large impact on learning robust features, also indicated by (Gavrikov & Keuper, 2022). Note, when $|A|$ is less than half the dataset, clean accuracy is improved over full AT (gray dot). In the appendix, section A.7, we show it is possible to trade-off these improvements for additional robust accuracy gains via TRADES Zhang et al. (2019).

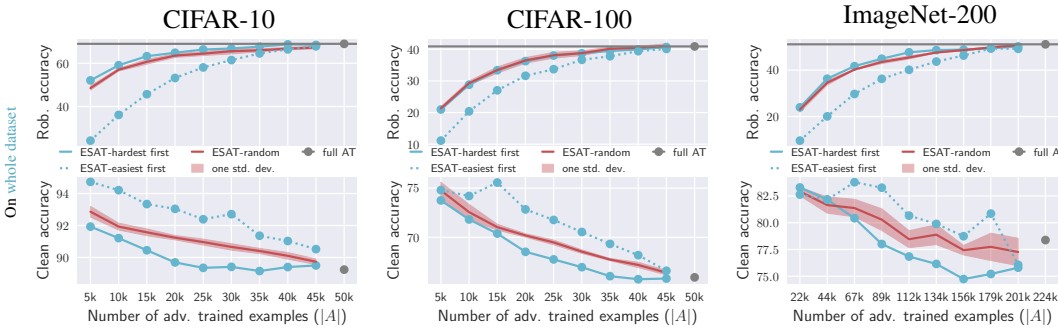

Figure 6: Example-subset Adversarial Training (ESAT) on CIFAR and ImageNet-200, provide quick convergence to a full AT baseline (gray line and dot) with increasing size of $A$. We report robust accuracy (upper row) and clean accuracy (lower row) and observe similar characteristics as with CSAT (figure 5). I.e., selecting the hardest examples first (solid line) provide higher rob. accuracy than easy ones (dashed line), although the gap substantially widens. Random example selection (red) provides competitive performance on average. Across all datasets, common clean accuracy decreases while robust accuracy increases (Tsipras et al., 2019). $L_\infty$ results in appendix, figure 18.

## 4.3 TRANSFER TO DOWNSTREAM TASKS

Previous experiments on ESAT could not provide explicit robust accuracies on the non-adv. trained subset $B_{val}$ since training and testing splits do not align naturally – recall the evaluation recipe outlined in section 3.3. In order to test transfer performance regardless, we make use of the fixed-feature task transfer setting proposed in (Shafahi et al., 2020). The recipe just slightly changes: split the data

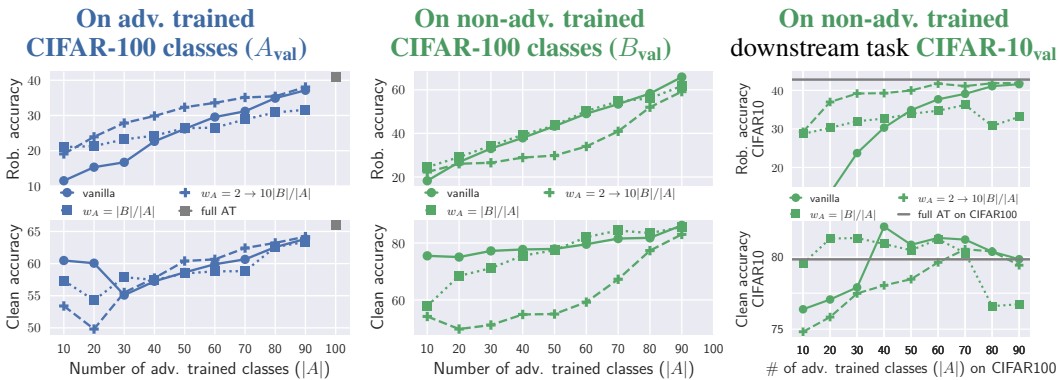

Figure 7: Impact of cross-entropy weighting on robustness transfer. For subset AT, we test different weighting strategies for sets A and B given they are of unequal size. We observe that vanilla cross-entropy (*circle*) offers the worst robustness transfer to CIFAR-10 (right). The best transfer (*plus*) is provided when loss weights are chosen such that training is overemphasized on A, indicated by dropping robust accuracies on B (compare left and center).

into $A$ and $B$ as usual and perform SAT. Fix all features, replace the last classification layer with a 1-hidden layered classifier and finetune only the new classifier on the target task. Importantly, neither training nor validation set for the target task are split. We consider CIFAR-100 and ImageNet-200 and transfer to CIFAR-10, SVHN, Caltech-256 (Griffin et al., 2007) and Flowers102 (Nilsback & Zisserman, 2008). We call SAT trained for transfer Source-task Subset Adversarial Training (S-SAT), to emphasize that the subset training is performed on the source-task dataset.

In this section, we consider models that have "seen" only a fraction of AEs on the source task and investigate the robustness transfer capabilities to tasks on which they have not explicitly adversarially trained on. We find unexpectedly strong transfer performances, boosted by putting more weight on the AEs.

**Loss balancing improves robustness transfer.** In contrast to the previously explored setting, we observe the transfer setting to benefit from loss balancing. Recall equation 2 in section 3.2 in which $w_A$ and $w_B$ can be assigned different values to balance the loss when $|A| \neq |B|$. We show that the vanilla configuration $w_A = w_B = 1$ transfers robustness to downstream tasks poorly, that balancing the loss with $w_B = 1, w_A = |B|/|A|$ lacks transfer performance for small $|B|$ and that weighting examples from $A$ higher results in improved robustness transfer. We present results for all three weightings in figure 7. The figure is organized in three columns, all reporting robust accuracy. The first column reports the robust accuracy on subset $A_{\text{val}}$, the second on subset $B_{\text{val}}$ and the third reports the robust accuracy on the downstream task. Here, we train on CIFAR-100 and transfer to CIFAR-10. The vanilla loss is indicated by circles and a solid line, the balanced loss $w_A = |B|/|A|$ by squares and a dotted line and the loss overemphasizing $A$ by a plus and a dashed line.

First and foremost, note that the robustness transfer for the vanilla configuration is substantially worse than both alternatives (robust accuracy in top right). Transfer improves with use of loss balancing, e.g. for $|A| = 10$, robust accuracy improves from 8% to 30%, but does not converge to the baseline AT performance (gray line). This is an unwanted side effect of equalizing the weight between $A$ and $B$. When $A$ is much smaller than $B$, less weight is assigned to the AEs constructed for $A$ and robustness reduces. Note, this effect can also be seen on $A_{\text{val}}$ (top left in figure). Instead, we find it beneficial to overemphasize on the AEs (plus with dashed line). This configuration assigns $w_A = 2|B|/|A|$ for $|A| = 10$ and increases the weight to $w_A = 10|B|/|A|$ for $|A| = 90$. This results in improved robust accuray on $A_{\text{val}}$, but low robust and clean accuracy on $B_{\text{val}}$. Interestingly, while the generalization to $B_{\text{val}}$ is low, robustness transfer to CIFAR-10 is very high. We use this loss weighting for all following task transfer experiments.

**Robustness transfer from example subsets.** Using the weighted loss, we focus in the following on S-ESAT on two source tasks: CIFAR-100 and ImageNet-200, and train on three downstream tasks. Similar results for S-CSAT and SVHN as additional downstream task can be found in the supplement, sections A.5 and A.9. Figure 8 presents results for three settings: CIFAR-100 → CIFAR-10 and ImageNet-200 → Caltech-256, Oxford-Flowers-102. The first and second row show robust and clean accuracy on the downstream task respectively. As before, we compare with a random (red) and a full

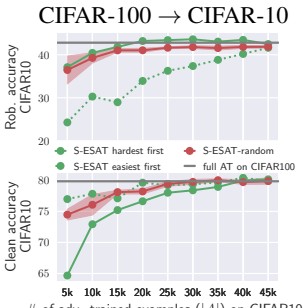 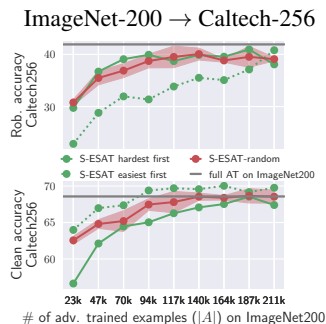 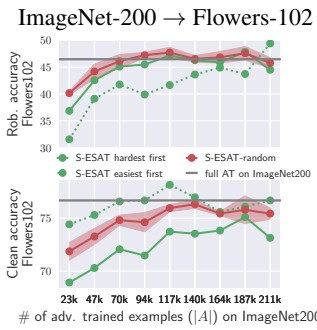

Figure 8: Transfer from S-ESAT to three different downstream tasks. S-ESAT is trained on source dataset CIFAR-100 (left) and ImageNet-200 (middle and right). We report robust (top row) and clean (bottom) accuracies for increasing size of $A$. Similar to our investigation on transfer from $A$ to $B$, we find that hard examples provide better robustness transfer than easy ones, but random selections (red) achieve competitive performances. Most importantly, "seeing" only few AEs (here 30% of source data) recovers baseline AT performance (gray line). $L_\infty$ results in appendix, figure 19.

AT baseline (gray line). Selecting $A$ to contain the hardest examples first (highest entropy) is marked by a solid line; selecting easiest is marked by a dashed line. Similar results for $L_\infty$ are provided in the appendix, section A.8.

In line with the improvements seen using appropriate loss weighting, we see similarly fast recovery of baseline AT performance across all dataset. In fact, $|A|$ containing only 30% of training data (15k and 70k) is sufficient to reach near baseline performance. On CIFAR-100 → CIFAR-10 and ImageNet-200 → Flowers-102 even slightly outperforming the same with a further increase in size. Similar to the non-transfer settings tested before, we see similar interactions between subset selection strategies. I.e. hardest examples (solid line) provide greater robustness transfer than easiest (dashed line) while a random baseline (red) achieves competitive performances. The latter consistently outperforming entropy selection on ImageNet-200 → Flowers-102, supporting our observation in section 4.2: with increasing dataset complexity, informed subset selection provides diminishing returns. Note that all robust accuracy gains correlate proportionally to an increase in clean accuracy as well. This is in stark contrast to the inverse relationship in previous settings. C.f. figure 5 and 6, for which clean accuracy decreases. This interaction during transfer is similar to what is reported in (Salman et al., 2020): increased robustness of the source model results in increased clean accuracy on the target task (over a non-robust model). Intriguingly though, with appropriate weighting, the biggest robustness gains on the downstream task happen under fairly small $A$. This is a promising outlook for introducing robustness in the foundational setting (Bommasani et al., 2021), where models are generally trained on very large datasets, for which AT is multiple factors more expensive to train. Note that our results generalize to single-step attacks like fast gradient sign method (FGSM) (Goodfellow et al., 2015; Wong et al., 2020) as well. We provide evaluations in the supplement, section A.10. While we consider the fixed-feature transfer only, recent work has shown this to be a reliable indicator for utility on full-network transfer (Salman et al., 2020; Kornblith et al., 2019).

## 5 CONCLUSION

We presented an analysis of how adversarial robustness transfers between classes, examples and tasks. To this end, we proposed the use of Subset Adversarial Training (SAT), which splits the training data into $A$ and $B$ and constructs AEs on $A$ only. Trained on CIFAR-10, CIFAR-100 and ImageNet-200, SAT revealed a surprising generalizability of robustness between subsets, which we found to be based on the following observations: (i) adv. robustness transfers among classes even if some or most classes have never been attacked during training and (ii) hard classes and examples provide better robustness transfer than easy ones. These observations remained largely valid in the transfer to downstream tasks like Flowers-102 and Caltech-256 for which we found that overemphasizing loss minimization of AEs in $A$ provided fast convergence to baseline AT robust accuracies, even though transfer to $B$ was severely reduced. Specifically, it appears that only few AEs ($A$ containing 30% of the training set) learn all of the robust features which generalize to downstream tasks. This finding could be particularly interesting for AT in the foundational setting, in which very large datasets make full AT challenging and 90% of the full robustness using 30% of data is preferred over no robustness.

**Acknowledgements** This work was partially funded by the ELSA – European Lighthouse on Secure and Safe AI funded by the European Union under grant agreement No. 101070617. Views and opinions expressed are however those of the authors only and do not necessarily reflect those of the European Union or European Commission. Neither the European Union nor the European Commission can be held responsible for them.

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

## A APPENDIX

### A.1 FULL TRAINING DETAILS

For all training setups listed in table 2, we train our models from scratch using SGD with a momentum of $0.9$. Dataset sizes are listed in table 1. All are data augmented based on the definitions in (Engstrom et al., 2019). The sequence of transformations are listed in figure 9. Left, for CIFAR-10, CIFAR-100 and SVHN. Right, for ImageNet-200, Caltech-256 and Flowers-102.

**Adversarial training** for the $L_2$ norm is performed with 7 steps of projected gradient descent (PGD-7) within an $\epsilon_2 = 0.5$ for CIFAR and SVHN and $\epsilon = 3.0$ for ImageNet-200, Caltech-256 and Flowers-102. For each step, we use a step size of $0.1$ and $0.5$ respectively. For the $L_\infty$ norm, we use 10 steps of PGD during ESAT and a step size of $1/255$ to avoid catastrophic forgetting when $|A|$ is small (see section A.8). For S-ESAT, we find it to be sufficient to perform PGD-7 with a step-size of $2/255$. This is likely mitigated by the weighted loss. For all datasets, we constrain the maximum perturbation norm to $\epsilon_\infty = 8/255$. For all experiments, including $L_2$ and $L_\infty$, we maximize the default cross-entropy loss. For vanilla AT, all examples in a training batch are attacked. For all SAT variants, we randomly sample training examples to construct a training batch and attack only examples that are contained in $A$.

**Class order.** In the following, we list the order of classes ranked by entropy $\overline{\mathcal{H}}_C$ (equation 3). CIFAR-10 can be derived from figure 4 in the main paper. In figures 10 and 11, we provide the list for CIFAR-100 and ImageNet-200. On CIFAR-100, the first and thus hardest classes consist mostly of animate categories like *otter*, *rabbit* and *crocodile*. The easiest on the other hand are inanimate categories, specifically vehicle related classes, e.g. *road*, *motorcycle* or *pickup-truck*. Overall, the animate-inanimate order is similar to CIFAR-10. On ImageNet-200, we observe a very different order. Inanimate categories like *spatula*, *drumstick* or *umbrella* are among the hardest, while animate classes like *monarch (butterfly)*, *flamingo* or *lorikeet* are among the easiest. Named hard classes may be difficult to distinguish due to a frequent presence of people in the images.

### A.2 ALTERNATIVE RANKINGS

For simplicity, we focused our experiments on using entropy as a proxy to measure example and class difficulty (c.f. equation 3). Multiple such difficulty metrics have been proposed in literature (Chang et al., 2017; Hua et al., 2021; Baldock et al., 2021; Agarwal et al., 2022), of which we select a few from recent literature to compare to: signed variance (SVar) (Hua et al., 2021) and variance of gradients (VoG) (Agarwal et al., 2022). We want to highlight, that they perform very similar to our entropy metric when utilized in our SAT framework. Figure 12 compares these two metrics with our used entropy metric using ESAT on CIFAR-100. Overall, VoG has a slight edge over SVar and Entropy, yet the differences remain small. On $5k$ attacked examples, Entropy (yellow line) achieves $21.0\%$, VoG (red line) $21.9\%$ and SVar (purple line) $22.3\%$ robust accuracy. On $25k$ attacked examples, Entropy achieves $38.0\%$, VoG $38.8\%$ and SVar $38.1\%$. While some improvements over our simple Entropy metric are possible, no proposed metric has a clear edge over the other.

```
 - pad 4 pixels
 - random crop to 32x32          - random crop to 224x224
 - random horizontal flip        - random horizontal flip
 - color jitter [0.25, 0.25, 0.25]  - color jitter [0.1, 0.1, 0.1]
 - random rotation within +/- 2 deg. - random rotation within +/- 2 deg.
```

Figure 9: Input transformation for CIFAR and SVHN datasets (left) and ImageNet-200, Caltech-256 and Flowers-102 (right) during training. During testing, no transformations are applied to CIFAR and SVHN. The remaining datasets are resized such that the shortest side equals 256, after which they are center cropped to 224.

```
otter, lizard, seal, rabbit, mouse, crocodile, lobster, shrew, shark,
woman, beaver, bowl, turtle, squirrel, possum, snail, girl, kangaroo,
ray, forest, caterpillar, man, baby, dinosaur, lamp, elephant, couch, boy,
porcupine, snake, butterfly, leopard, crab, table, mushroom, dolphin,
willow_tree, beetle, spider, clock, fox, sweet_pepper, bee, house,
raccoon, tulip, bridge, bus, rose, tank, whale, train, worm, lion, poppy,
trout, bed, plate, can, telephone, tiger, hamster, aquarium_fish,
maple_tree, orchid, pear, mountain, tractor, oak_tree, rocket, skunk,
cockroach, television, cup, sea, cloud, lawn_mower, castle, bottle,
palm_tree, keyboard, apple, plain, pickup_truck, bicycle, orange, chair,
wardrobe, motorcycle, road
```

Figure 10: CIFAR-100 classes ranked by decreasing entropy $\overline{\mathcal{H}}_C$. Animal classes are hardest, inanimate classes easiest.

```
spatula, shovel, syringe, drumstick, hand blower, lighter, nail, maraca,
barrow, umbrella, bow, quill, iron, stethoscope, soap dispenser, dumbbell,
mask, reel, toaster, ant, walking stick, envelope, candle, sleeping bag,
sandal, tricycle, cowboy boot, cradle, breastplate, bubble, banjo, chest,
cliff, wine bottle, fountain, crayfish, doormat, Chihuahua, chain, apron,
kimono, cockroach, accordion, sewing machine, ocarina, revolver, torch,
piggy bank, goblet, studio couch, wreck, hermit crab, grand piano, beaker,
snail, marimba, sundial, mantis, vulture, sea lion, flagpole, washer,
acoustic guitar, mongoose, grasshopper, Christmas stocking, bikini, corn,
balance beam, fox squirrel, American alligator, academic gown,
feather boa, suspension bridge, stingray, acorn, common iguana, forklift,
parachute, mushroom, hotdog, American black bear, beacon, garbage truck,
cello, pug, bee, banana, volcano, baboon, centipede, golfcart, marmot,
limousine, African chameleon, leafhopper, canoe, wood rabbit, agama,
starfish, lynx, German shepherd, capuchin, balloon, goose,
submarine, golden retriever, mitten, jeep, hummingbird, armadillo,
weevil, porcupine, puck, snowplow, barn, fly, tarantula, Rottweiler,
pool table, red fox, harvestman, pretzel, ballplayer, American egret,
puffer, ladybug, pelican, obelisk, bald eagle, go-kart, bell pepper,
castle, snowmobile, junco, lemon, spider web, lion, water tower,
basketball, guacamole, toucan, tank, jellyfish, viaduct,
robin, ambulance, broccoli, flatworm, pomegranate, bison, sea anemone,
jay, rugby ball, organ, drake, cheeseburger, mosque, koala, garter snake,
African elephant, lycaenid, oystercatcher, box turtle, cabbage butterfly,
steam locomotive, goldfinch, jack-o'-lantern, school bus, lorikeet,
manhole cover, rapeseed, flamingo, yellow lady's slipper, monarch
```

Figure 11: ImageNet-200 classes ranked by decreasing entropy $\overline{\mathcal{H}}_C$. In contrast to the order on CIFAR-10 and CIFAR-100, animate classes are generally not the most frequent among the hardest. Instead its mostly inanimate objects.

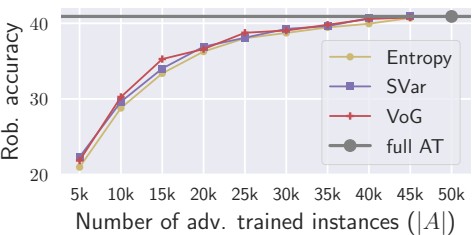

Figure 12: Various hardness metrics result in similar rob. accs. for ESAT on CIFAR-100.

## A.3 Single class CSAT with different $\epsilon$

Supplementary to our CIFAR-10 single class results in figure 2, we provide additional results for smaller and larger $L_2$ $\epsilon$ in figure 13. Overall, we continue to observe non-trivial robustness transfer to $B$, irrespective of $\epsilon$. Interesting though, the robustness transfer notably increases for smaller $\epsilon = 0.25$ (left), but also notably decreases for larger $\epsilon = 1.0$ (right).

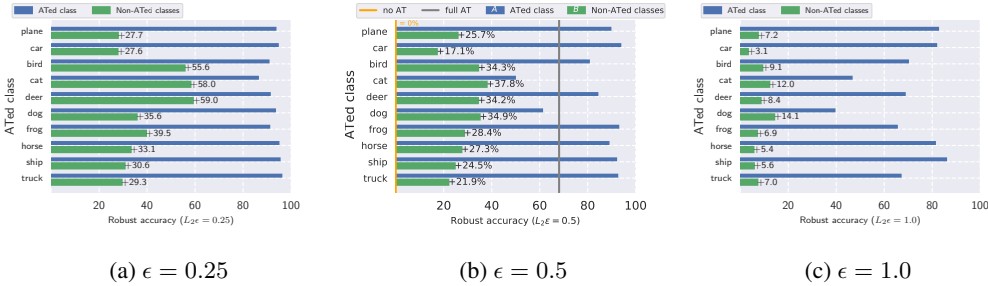

(a) $\epsilon = 0.25$        (b) $\epsilon = 0.5$        (c) $\epsilon = 1.0$

Figure 13: For all $\epsilon$, we observe non trivial robustness transfer to $B$, yet transfer diminishes with increasing $\epsilon$. For smaller $\epsilon$ though, the transfer is notably stronger. For $\epsilon = 1.0$ we use a step-size of 0.2 and 0.1 otherwise.

## A.4 Dataset augmentation – CIFAR-10 +1

As briefly mentioned in the main paper, we can utilize the class-entropy to robustness transfer relationship to augment a dataset by samples with high entropy. Then, SAT is performed only on these new samples to establish strong robustness on the original dataset. As a proof-of-concept, we synthesized a set of 11th classes from CIFAR-100s super-classes (see figure 15) – of which there are 20 – and perform SAT on this 11th class to evaluate the robust accuracy gains on the original CIFAR-10 classes. Results are reported in figure 14. We continue to observe a correspondence between average entropy $\mathcal{H}_c$ and the robustness transfer of a class. As in the main paper, we evaluate $\mathcal{H}_c$ on non-adv. trained models. Note that, while the best performing setup ($A = \{rodent\}$) with rob. acc of 33.3% does not improve upon the best in the main paper ($A = \{cat\}$, with a rob. acc $> 37.8\%$), the number of examples in $A$ is only $|A| = 2500$, thus less than 5% of training data. This provides an indication that such a dataset augmentation is possible.

## A.5 Full results for CSAT

Results for CSAT can be plotted for three different validation subsets: $A_{\text{val}}$, $B_{\text{val}}$ and on the whole dataset $\mathcal{D}_{\text{val}}$. For clarity, we only showed robust accuracies on $\mathcal{D}_{\text{val}}$ and $B_{\text{val}}$ in the main paper in figure 5. Here, we provide all results. That is, in figure 16, we show robust accuracies in the upper split and clean accuracies in the lower split for all 3 subsets.

| Dataset | Classes | Size (Train/Test) |
|---|---|---|
| CIFAR-10 (Krizhevsky et al., 2009) | 10 | 50 000 / 10 000 |
| CIFAR-100 (Krizhevsky et al., 2009) | 100 | 50 000 / 10 000 |
| ImageNet-200 (Deng et al., 2009; Hendrycks et al., 2021) | 200 | 259 906 / 10 000 |
| Caltech-256 (Griffin et al., 2007) | 257 | 24 485 / 6122 |
| Flowers-102 (Nilsback & Zisserman, 2008) | 102 | 1020 / 1020 |
| SVHN (Netzer et al., 2011) | 10 | 73 257 / 26 032 |

Table 1: Number of training and validation examples per dataset used. ImageNet-200 uses examples from (Deng et al., 2009) only for classes defined in (Hendrycks et al., 2021)

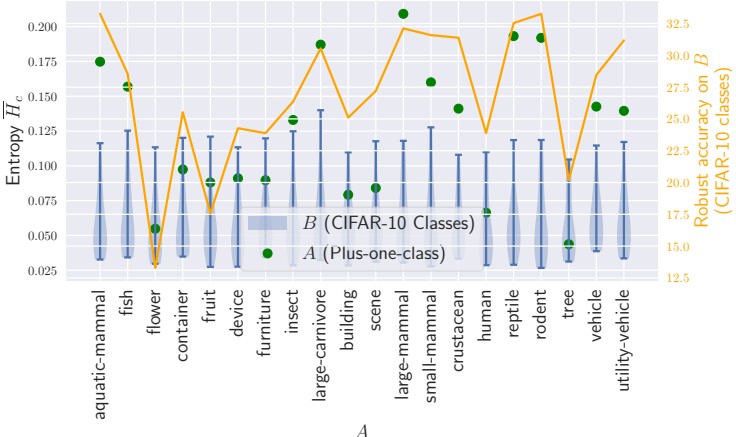

Figure 14: CSAT on CIFAR10 plus an additional class, synthesized from CIFAR-100. $A$ consists only of this additional 11th class, to test how much robust accuracy can be gained on the original CIFAR-10 classes (orange). We observe a consistent link with the entropy of this 11th class (green) with respect to the entropy of the CIFAR-10 classes (blue). The higher $\mathcal{H}_c$, the higher the robust accuracy.

## A.6 CLEAN ACCURACY INDEPENDENT CSAT RESULTS

So far, we presented robust accuracies as well clean accuracies. To highlight, that the hardest examples provide strongest robustness transfer which is independent of clean accuracy gains, we show in figure 17 an additional robustness metric: *Attack defense rate*. We define *Attack defense rate* as the robust accuracy on purely accurately classified examples. That is, given a dataset, we select examples that are accurately predicted and perform AA. The fraction of robust examples are reported. We observe, that the relationship between hardest, easiest and random examples are retained w.r.t. figure 16 on all dataset. We thus conclude that the reported gains are independent of clean accuracy.

## A.7 ROBUSTNESS ACCURACY TRADE-OFF WITH TRADES

In the following, we investigate the impact of applying *TRADES* (Zhang et al., 2019) – adversarial training with tuneable trade-off between clean and robust accuracy. This is of particular interest given

| Conventional setting | | | | | | |
|---|---|---|---|---|---|---|
| **Dataset** | **Architecture** | **Epochs** | **Batchsize** | **lr** | **lr-decays** | **$L_2$ decay** |
| CIFAR-10 | PreActResNet-18 | 200 | 128 | 0.1 | 100, 150 | $5 \cdot 10^{-4}$ |
| CIFAR-100 | PreActResNet-18 | 200 | 128 | 0.1 | 100, 150 | $5 \cdot 10^{-4}$ |
| ImageNet-200 | ResNet-50 | 150 | 256 | 0.1 | 50, 100 | $1 \cdot 10^{-4}$ |
| Transfer setting | | | | | | |
| **Dataset** | **Architecture** | **Epochs** | **Batchsize** | **lr** | **lr-decays** | **$L_2$ decay** |
| CIFAR-10 | PreActResNet-18 + [512,10] | 40 | 128 | 0.1 | 20, 30 | $5 \cdot 10^{-4}$ |
| SVHN | PreActResNet-18 + [512,10] | 40 | 128 | 0.1 | 20, 30 | $5 \cdot 10^{-4}$ |
| Caltech-256 | ResNet-50 + [2048,257] | 100 | 128 | 0.1 | 50, 75 | $1 \cdot 10^{-4}$ |
| Flowers-102 | ResNet-50 + [2048,102] | 100 | 102 | 0.1 | 50, 75 | $1 \cdot 10^{-4}$ |

Table 2: Training settings for all used datasets for the conventional (upper rows) and the transfer setting (lower rows). In the transfer setting, the last classifier layer is replaced with two linear layers of size $K \times K$ and $K \times N$, abbreviated as $[K, N]$. $K$ defines the number of feature channels and $N$ the number of classes.

```
aquatic-mammal: ['beaver', 'dolphin', 'otter', 'seal', 'whale']
fish: ['aquarium_fish', 'flatfish', 'ray', 'shark', 'trout']
flower: ['orchid', 'poppy', 'rose', 'sunflower', 'tulip']
container: ['bottle', 'bowl', 'can', 'cup', 'plate']
fruit: ['apple', 'mushroom', 'orange', 'pear', 'sweet_pepper']
device: ['clock', 'keyboard', 'lamp', 'telephone', 'television']
furniture: ['bed', 'chair', 'couch', 'table', 'wardrobe']
insect: ['bee', 'beetle', 'butterfly', 'caterpillar', 'cockroach']
large-carnivore: ['bear', 'leopard', 'lion', 'tiger', 'wolf']
building: ['bridge', 'castle', 'house', 'road', 'skyscraper']
scene: ['cloud', 'forest', 'mountain', 'plain', 'sea']
large-mammal: ['camel', 'cattle', 'chimpanzee', 'elephant', 'kangaroo']
small-mammal: ['fox', 'porcupine', 'possum', 'raccoon', 'skunk']
crustacean: ['crab', 'lobster', 'snail', 'spider', 'worm']
human: ['baby', 'boy', 'girl', 'man', 'woman']
reptile: ['crocodile', 'dinosaur', 'lizard', 'snake', 'turtle']
rodent: ['hamster', 'mouse', 'rabbit', 'shrew', 'squirrel']
tree: ['maple_tree', 'oak_tree', 'palm_tree', 'pine_tree', 'willow_tree']
vehicle: ['bicycle', 'bus', 'motorcycle', 'pickup_truck', 'train']
utility-vehicle: ['lawn_mower', 'rocket', 'streetcar', 'tank', 'tractor']
```

Figure 15: 20 CIFAR-100 Superclasses for reference.

that clean accuracy for small $|A|$ is higher than with full AT (e.g. see figure 6, enabling improved robust accuracy.

| $A = \{\text{cat}\}$ | $\beta$ | clean acc. in % on CIFAR-10 | robust acc. in % on $A$ | on $B$ | on CIFAR-10 |
|---|---|---|---|---|---|
| w/o TRADES | N/A | 91.0 | 49.6 | 37.8 | 39.0 |
| w/ TRADES | 1.0 | 92.4 | 41.5 | 32.0 | 33.0 |
| | 6.0 | 85.9 | 70.2 | 42.1 | 44.9 |
| | 12.0 | **81.0** | **78.2** | **42.9** | **46.4** |
| | 24.0 | 81.6 | 80.1 | 41.8 | 45.6 |

Table 3: TRADES (Zhang et al., 2019), applied to CSAT improves robustness transfer from single class *cat* to $B$ over baseline CSAT training. That is, we gain $5.1\%$-points to achieve a robust accuracy on $B$ of $42.9\%$.

| **TRADES on ESAT** | $|A| =$ | **5k** | **10k** | **15k** | **20k** | **25k** | **30k** | **35k** | **40k** | **45k** | **full AT** |
|---|---|---|---|---|---|---|---|---|---|---|---|
| | $\beta =$ | 12.0 | 11.25 | 10.5 | 9.75 | 9.0 | 8.25 | 7.5 | 6.75 | 6.0 | 6.0 |
| yes | Rob. acc | 57.2 | 62.1 | 65.7 | 66.1 | 67.7 | 67.6 | 68.2 | 66.4 | 69.0 | 69.0 |
| | Clean acc | 87.0 | 87.4 | 84.2 | 85.6 | 84.3 | 87.0 | 84.0 | 88.0 | 88.4 | 88.6 |
| no | Rob. acc | 52.1 | 59.1 | 63.3 | 64.9 | 66.3 | 66.8 | 67.7 | 68.7 | 68.5 | 69.0 |
| | Clean acc | 91.9 | 91.2 | 90.5 | 89.7 | 89.4 | 89.4 | 89.2 | 89.4 | 89.5 | 89.2 |

Table 4: ESAT on CIFAR-10 combined with TRADES (Zhang et al., 2019). Trading off clean for robust accuracy, we gain $\geq 66.0\%$ robust accuracy earlier at $20k$ samples in contrast to $25k$ for vanilla ESAT. However, our $\beta$ choice decreases clean accuracy below full AT.

Zhang et al. (2019) proposed a loss with principled trade-off capability between clean and robust accuracy for AT. We observe that clean accuracy is higher than with full AT when $|A|$ is small, i.e. less than $50\%$ of examples. We investigate to what extent robust accuracy can be improved. First, we conduct a hyperparameter sweep for TRADES $\beta$ parameter which controls the trade-off. As baseline we choose the best performing CSAT configuration on CIFAR-10 with $L_2\epsilon = 0.5$, that is $A = \{\text{cat}\}$. Results are presented in table 3. Note that – as expected – clean accuracy decreases with increasing $\beta$ while robust accuracy increases. Although Zhang et al. (2019) recommend $\beta$ to be less than 10.0, we find 12.0 to provide best robustness transfer from $A$ to $B$. This configuration achieves $42.9\%(+5.1)$ robust accuracy on $B$ while also increasing rob. acc. on $A$ to $78.2\%(+28.6)$, with the expected decrease in overall clean accuracy $(81.0\%(-10.0))$.

In a second experiment, we apply TRADES to ESAT on CIFAR-10. Given the $\beta = 6.0$ recommendation in (Zhang et al., 2019) and our $\beta = 12.0$ observation for single class CSAT, we use linear interpolation between these two values for general ESAT. That is, when $|A|$ is small, we trade off some of the gained clean accuracy for increased robustness with large $\beta = 12$, but decrease to $\beta = 6$

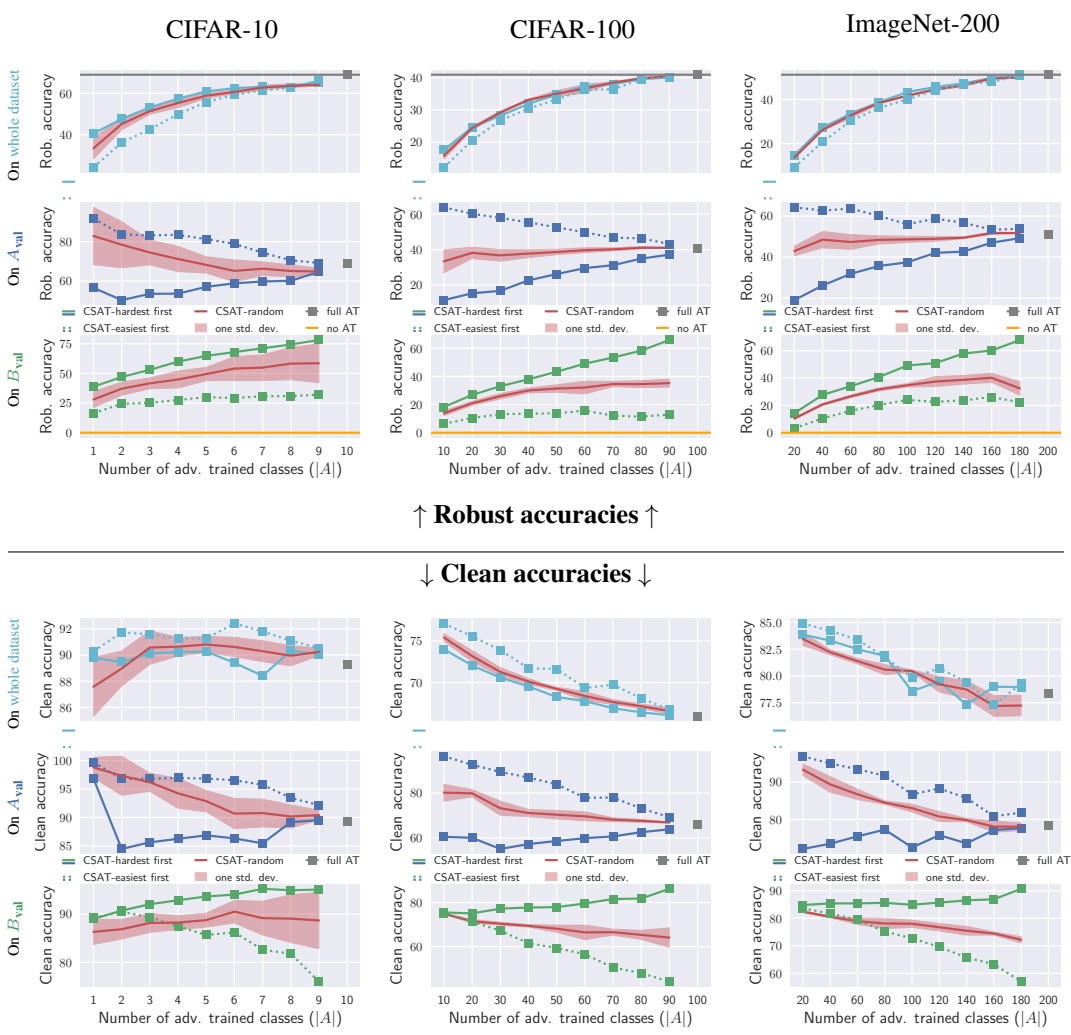

CIFAR-10    CIFAR-100    ImageNet-200

↑ **Robust accuracies** ↑

↓ **Clean accuracies** ↓

Figure 16: Full robust (upper split) and clean accuracies (lower split) from CSAT experiments, plotted for the whole dataset, $A_{\text{val}}$ and $B_{\text{val}}$. Selecting the hardest classes first (solid lines), clean accuracies and robust accuracies on $A_{\text{val}}$ steadily increase, while selecting the easiest in contrast (dotted lines) results in a steady decline. This provides additional support that entropy as metric provides a useful account of difficulty, since easy classes can achieve higher accuracy. Furthermore, we note that clean accuracy on the whole dataset is increasing or mostly stable, while on other datasets it is steadily decreasing. This should be investigated further.

with increasing size of $|A|$. We compare *ESAT-hardest first* with and without TRADES in table 4. We observe quicker robust accuracy convergence to the baseline. $66\%$ robust accuracy is achieved at $20k$ samples, instead of at $25k$. However, our choice of $\beta$ also induces a drop in clean accuracy below full AT clean accuracy ($85.6\%$ clean accuracy at $20k$ vs $89.7\%$ at $25k$ vs $89.2\%$ for full AT).

## A.8  $L_{\infty}$ RESULTS

While the main paper focuses on the $L_2$ norm, we also provide a corresponding evaluation for $L_{\infty}$.

**ESAT.** We evaluate ESAT on CIFAR-10, CIFAR-100 and ImageNet-200 in figure 18 and S-ESAT on CIFAR-100 → CIFAR-10 and ImageNet-200 → {Caltech-256, Flowers-102}. For ESAT in figure 18, we find characteristics similar to the $L_2$ results in figure 6. That is, few examples contribute a large amount of overall robustness to the model and robustness increases quickly with size of $A$.

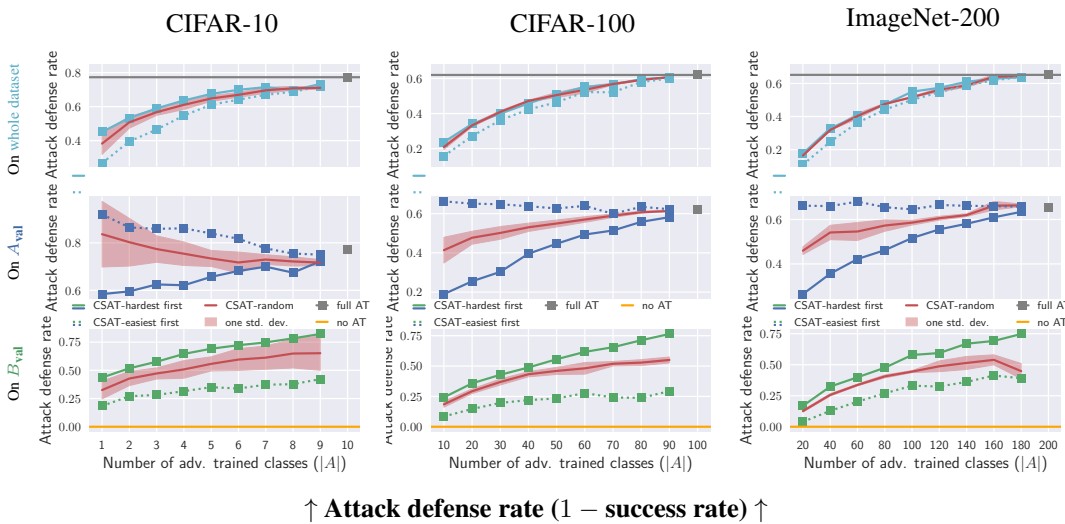

$$\uparrow \textbf{Attack defense rate } (1 - \textbf{success rate}) \uparrow$$

Figure 17: As complement to the CSAT results in figure 16, we additionally show the defense rates (for what fraction of accurately classified validation examples does Auto-Attack fail to find adv. examples). As the defense rate shows robustness independent of clean accuracy, we can test whether the robustness gains are in fact based on the improved robustness transfer from hard classes. We see this being the case across all datasets.

Noteworthy though, is the low robust accuracy when $A$ contains $10\%$ of data, dropping to $0\%$ on CIFAR-10 and CIFAR-100. We have found this to be similar to catastrophic overfitting described for single-step AT (Wong et al., 2020). The effect can be mitigated with more PGD iterations (10) and a smaller step size ($1/255$), but is not resolved entirely. That is, note that the drop only occurs for the hardest examples, but not for the easiest or a random selection.

**S-ESAT.** For S-ESAT, we show the same transfer configurations as in the main paper. In figure 19, we observe similar characteristics as for $L_2$. That is, fast convergence to baseline AT performance and hard examples providing better robust transfer than easy examples. Also similar to $L_2$, is the observation that robustness transfer from ImageNet-200 to Flowers-102 is best when using a random selection of examples.

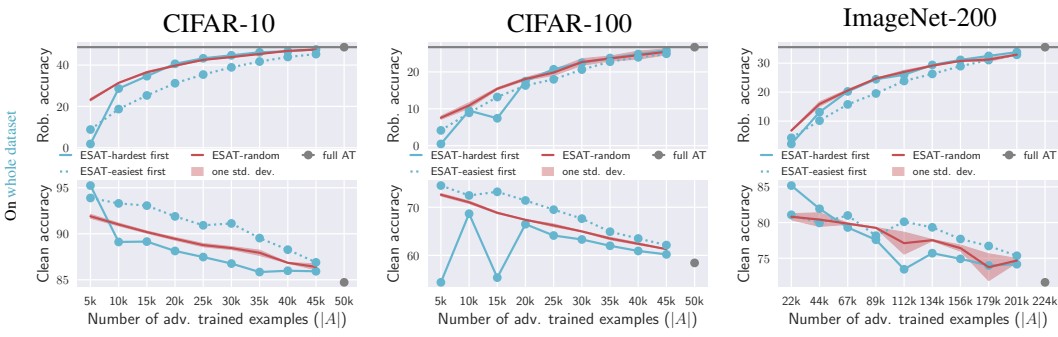

Figure 18: Example-subset Adversarial Training (ESAT) on CIFAR datasets and ImageNet-200 using $L_\infty$ with norm $\epsilon_\infty = 8/255$, provide quick convergence to a full AT baseline (gray line and dot) with increasing size of $A$. We report robust accuracy (upper row) and clean accuracy (lower row) and observe similar characteristics as with CSAT (figure 5). I.e., selecting the hardest examples first (solid line) provide higher rob. accuracy than easy ones (dashed line), although the gap substantially widens. Random example selection (red) provides competitive performance on average. Across all datasets, we see the common clean accuracy decrease while robust accuracy increases (Tsipras et al., 2019).

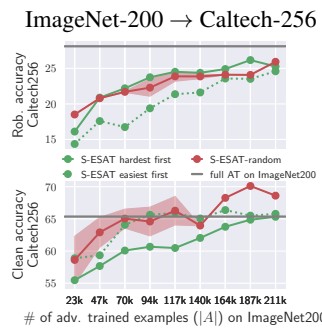

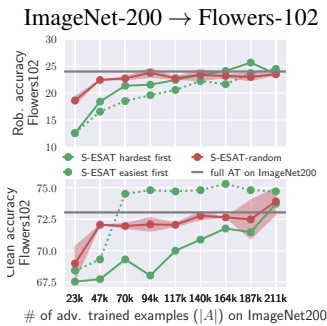

Figure 19: Transfer from S-ESAT to three different downstream tasks using $L_\infty$ with norm $\epsilon_\infty = {}^8\!/_{255}$. S-ESAT is trained on source dataset CIFAR-100 (left) and ImageNet-200 (middle and right). We report robust (top row) and clean (bottom) accuracies for increasing size of $A$. Similar to our observations for $L_2$ in figure 8, we find that hard examples provide better robustness transfer than easy ones, but random selections (red) achieve competitive performances. Most importantly, "seeing" only few AEs (here 30% of source data) recovers baseline AT performance (gray line).

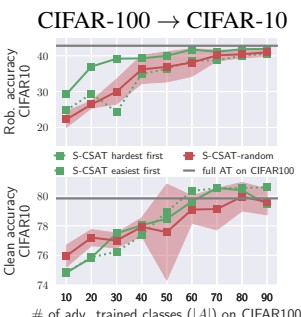

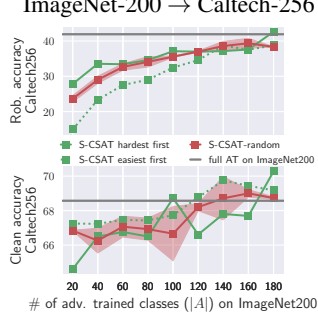

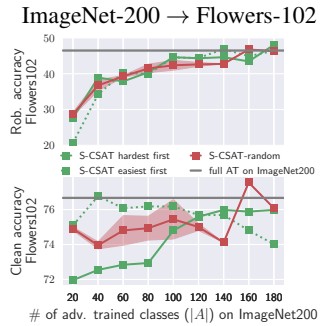

Figure 21: Transfer from S-CSAT to the same downstream tasks as in figure 8. S-CSAT is trained on source dataset CIFAR-100 (left) and ImageNet-200 (middle and right). We report robust (top row) and clean (bottom row) accuracies for increasing size of $A$. We observe similar properties to S-ESAT, yet find convergence to the baseline AT performance to be substantially slower; in line with our discussion on SAT in section 3.2.

## A.9 FULL RESULTS FOR TRANSFER SETTINGS

In the main paper, we omitted transfer results to SVHN as well as using S-CSAT. Firstly, we provide the transfer result from CIFAR-100 to SVHN in figure 20. Robust accuracies are plotted on the upper plot, clean accuracies below. Note that $5k$ examples in $A$ are sufficient to reach baseline AT performance (gray line), while $15k$ provides a substantial improvement in robust accuracy ($\sim 22\%$ vs $\sim 20\%$). Secondly, transfer results on S-CSAT aligned with the experiments in section 4.3 are shown in figure 21. We observe similar characteristics to the CSAT results in section 3.2, i.e. selecting the hardest classes first (solid line) is only advantageous on small $A$, while generally it draws even with the random baseline (red). Overall, convergence to the full AT baseline is slower than with S-ESAT.

## A.10 SINGLE-STEP AT

While our main $L_2$ experiments use AT with 7 PGD-steps, we here show that non-trivial robustness transfer can be achieved with single-step AT as well. We focus on transfer to downstream

Figure 20: Robustness transfer from CIFAR-100 to SVHN using S-ESAT

tasks and compare with the results shown in figure 8, section 4.3. I.e., we train one ESAT model on CIFAR-100 and ImageNet-200 respectively, and finetune an additional classifier on either CIFAR-10, Caltech256 or Flowers-102. We use *FGSM-RS* (Wong et al., 2020), with a step-size of $0.625$ for $\epsilon = 0.5$ and $3.75$ for $\epsilon = 3.0$. All other training settings are consistent with previous experiments (c.f. section A.1).

Results are shown in figure 22, comparing PGD-7 training (circles on solid line) and single-step FGSM-RS (squares on dotted line). Generally, we observe very similar clean and robust accuracies (lower and upper row) across all architectures. Specifically, FGSM-RS achieves slightly higher clean accuracies and slightly lower robust accuracies – especially for small $|A|$. Nonetheless, single-step AT converges to the full AT baseline (gray line) in a similar fast rate, i.e. generating AEs for around $30\%$ of the training set is sufficient.

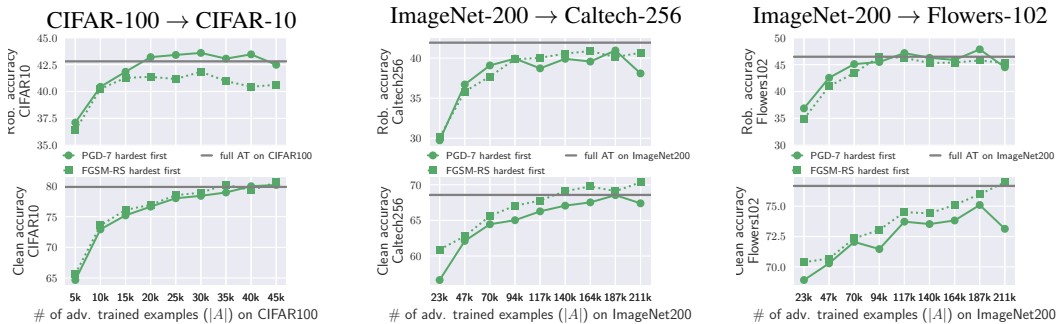

Figure 22: Comparison between PGD-7 and single step S-ESAT on the transfer setting to three different downstream tasks. Training and evaluation using $L_2$ norm. Our observations on robustness transfer remain valid even for single step attacks. Except for CIFAR-100 $\rightarrow$ CIFAR-10, we find robust accuracies and clean accuracies to be consistent with PGD-7.

