# OpenReview forum: "On Adversarial Training without Perturbing all Examples"
_ICLR.cc/2024/Conference — ICLR 2024 poster_

### Official Review · Reviewer_uHfk · 2023-10-15

**Soundness:** 3 good
**Presentation:** 2 fair
**Contribution:** 3 good
**Rating:** 6
**Confidence:** 4

**Summary:**

In this paper, the authors study the generalization of adversarial robustness from class-wise and sample-wise aspects.

**Strengths:**

1. There are two different settings, i.e., CSAT and ESAT. For each of them, there are comprehensive experiments based on the proposed entropy metrics. On the other hand, the authors consider both $L_2$ and $L_\infty$ attacks. All settings are studied on multiple datasets, which makes the results more convincing.

2. Downstream task transferability is an interesting topic. The results indicate that the learned features can be transferred to other classes, which is aligned with the observation from CSAT.

3. This paper is easy to follow. The writing is clear.

**Weaknesses:**

1. This paper mainly contains various experiments and their results, but lack the important analysis. Specifically, there are no analysis of the transferability observed from CSAT and ESAT. I cannot get any insightful information after reading this paper, although the results are informative.

2. There is no theoretical analysis to rethink the observation as a special property of adversarial training. Additionally, from Figure 4, we can find that the clean accuracy has a similar tendency as the robust accuracy, therefore, it is possible that CSAT and ESAT are just because of the generalizability of the deep learning models.

3. Only PGD-AT is considered. More advanced methods, like TRADES and AWP, should be evaluated under the same settings.

4. For downstream task transferability, it is similar, but not exactly the same, as contrastive adversarial training. The authors should discuss the similarity and difference between these two methods in this case.


Minor:

a. some figures' labels are blocked.

**Questions:**

1. I notice that the authors use stronger data augmentation than usual. For example, when training on CIFAR-10, we usually only use random crop and flip. I hope the authors can provide ablation studies on these data augmentation methods.

---

> ### Author Response · Authors · 2023-11-19
> **Question 1 to 3**
>
> 1. **no analysis:**
> We agree with the reviewer, that an additional analysis regarding the 'why' would be of interest. At this point, we cannot offer one but refer the reviewer to their own statement that the "*results are informative*". We also believe that our results are informative and are likely to spur follow up work. We believe this is a strong reason to publish this work.
> We restate the contribution of our paper: in contrast to related work on robustness and its transfer capabilities (as discussed in related work and our method section), we conduct a systematic analysis where we perturb only a fixed subset of training examples. This reveals a surprising phenomenon: adv. training a single class, transfers robustness to all other classes above non-trivial levels. Moreover, the hardest class provides the best transfer. As a consequence, we observe that just perturbing $50$% of training examples is sufficient to reach baseline AT performance and only $30$% of examples is sufficient in the task transfer setting.
>
> 2.
>     a) **no theory:** While additional theory is useful, we think that a lack of it does not reduce the importance of our work. Our work describes a phenomenon of adversarial training --few attacked examples/classes transfer robustness surprisingly well-- comprehensively.
>     This is confirmed by the reviewer itself and also R3h75, RgZFF and RQzqJ.
>
>     b) **effect is just side-effect of generalizability:** We thank the reviewer for a highly relevant question that is addressed by providing additional figures in the new paper revision reporting the defense rate -- robustness independent of clean accuracy. While the trend between clean and robust accuracy seems highly correlative, in section A.6 and figure 17 in the appendix, we show that the increase in robustness is not due to clean accuracy alone. By providing robustness on accurately classified examples only, we remove the influence of clean accuracy entirely. We can see that robustness transfer is best when SAT is performed using the hardest examples.
>
> 3. **no TRADES and AWP evaluation:** We agree with the reviewer that integration of TRADES is especially interesting in our setting, since it provides a means to trade-off clean with robust accuracy. However, we highlight that selecting the best trade-off between clean and robust accuracy with TRADES does not alter the main findings of our paper: adv. training on few examples/classes transfers robustness surprisingly well. Moreover, adversarial weight perturbation would be interesting to investigate, yet we believe that it goes beyond the scope of this work. Regarding TRADES, we have added results for CIFAR-10 for CSAT $L_2 \epsilon=0.5$ $|A|=1$ and the general case. For $|A|=1$ we conducted a coarse hyperparameter search for TRADES $\beta$ parameter and found $\beta=12.0$ to provide good robustness transfer (see table below). This configuration achieves $42.9$% $(+10.7)$ robust accuracy on $B$ while also increasing rob. acc. on $A$ to $78.2$% $(+28.6)$, with the expected decrease in overall clean accuracy ($81.0$% $(-10.2)$).
>
> |$A=\{cat\}$|$\beta$|clean acc on CIFAR-10|rob. acc on $A$|rob. acc on $B$|
> |--|--|--|--|--|
> |w/o TRADES|N/A|91.0|49.6|37.8|
> |w/ TRADES|1.0|92.4|41.5|32.0|
> ||6.0|85.9|70.2|42.1|
> ||12.0|**81.0**|**78.2**|**42.9**|
> ||24.0|81.6|80.1|41.8|
>
> In a second experiment, we use linearly decreasing $\beta$, from $12$ down to $6$, for our ESAT $L_2 \epsilon=0.5$ experiment on CIFAR-10, corresponding to figure 6.
> That is, when $|A|$ is small, we trade off some of the gained clean accuracy for increased robustness with large $\beta=12$, but decrease to $\beta=6$ with increasing size of $|A|$. We choose $\beta=6.0$ for $|A|=9$ based on the same parameter choice for the full CIFAR-10 set in (Zhang et al. 2019). We compare *ESAT-hardest first* with and without TRADES in the table below.
> We observe quicker robust accuracy convergence to the baseline ($66$% robust accuracy is achieved at $20k$ samples, instead of at $25k$) with an expected drop in clean accuracy below full AT baseline clean accuracy ($85.6$% clean accuracy at $20k$ vs $89.7$% at $25k$ vs $89.2$% for full AT).
>
>
> |TRADES||5k|10k|15k|20k|25k|30k|35k|40k|45k|full AT|
> |--|--|--|--|--|--|--|--|--|--|--|--|
> |yes|Rob. acc|57.2|62.1|65.7|66.1|67.7|67.6|68.2|66.4|69.0|69.0|
> ||Clean acc|87.0|87.4|84.2|85.6|84.3|87.0|84.0|88.0|88.4|88.6|
> |no|Rob. acc|52.1|59.1|63.3|64.9|66.3|66.8|67.7|68.7|68.5|69.0|
> ||Clean acc|91.9|91.2|90.5|89.7|89.4|89.4|89.2|89.4|89.5|89.2|

---

> ### Author Response · Authors · 2023-11-19
> **Question 4-6**
>
> 4. **what is difference to contrastive AT?:** We base our discussion around (Kim et al.), in which the authors provide a self-supervised adversarial training method, which does not require explicit labels. We highlight, that our proposed analysis is complementary to this work. (Kim et al.) perturb *all* training examples, similar to the vanilla AT approach. Ours, instead, investigates the impact of robustness transfer when we perturb only a *fixed subset*. Our SAT analysis is therefore applicable to the proposed contrastive learning approach.
>
>     Kim, Minseon, Jihoon Tack, and Sung Ju Hwang. "Adversarial self-supervised contrastive learning." Advances in Neural Information Processing Systems 33 (2020).
>
> 5. **some figures' labels are blocked:** We thank the reviewer for raising this issue. We have fixed the blocked labels in all figures.
>
> 6. **data augmentation different to related work:** We adopted the adversarial training methodology from the GitHub repository *github.com/MadryLab/robustness*. Consequently, we use the same data augmentation as they did. That is, on CIFAR-10, the data augmentation involves cropping, flipping, color jittering and rotating (for details see *github.com/MadryLab/robustness/blob/master/robustness/data_augmentation.py*).
> Irrespectively, we repeated our CSAT $|A|=1$ experiments with the reviewers suggested data augmentation pipeline: only performing cropping and flipping. Results, reported in the table below, show similar results to our reported in our submission.
> We conclude that the data augmentation is not the reason for our observed robustness transfer.
>
> |data augmentation|class in $A$ | clean acc on CIFAR-10|rob. acc. on $A$|rob. acc. on $B$|
> |--|--|--|--|--|
> |crop,flip|plane|88.6|88.8|26.7|
> ||car|89.2|95.4|15.4|
> ||bird|84.1|80.6|34.4|
> ||cat|91.0|50.4|39.9|
> ||deer|84.6|85.5|34.5|
> ||dog|91.5|61.2|35.0|
> ||frog|86.8|90.2|28.3|
> ||horse|88.2|90.2|27.6|
> ||ship|89.0|91.2|24.0|
> ||truck|91.9|87.0|20.3|
> ||
> |crop,flip,jitter,rotate|plane|87.2|89.4|25.7|
> ||car|89.8|93.5|17.1|
> ||bird|84.4|80.4|34.3|
> ||cat|91.0|49.6|37.8|
> ||deer|85.4|84.0|34.2|
> ||dog|91.5|60.9|34.9|
> ||frog|85.7|92.7|28.4|
> ||horse|88.1|88.6|27.3|
> ||ship|89.0|91.8|24.5|
> ||truck|87.6|92.3|21.9|

---

> ### Comment · Reviewer_uHfk · 2023-11-20
>
> Thanks for the response of the authors. I have read other reviewers' comments and responses. There are some additional questions.
>
> 1. The authors mention that the proposed method can reduce the time cost. Could the authors discuss related works, such as Free-AT[1]?
>
> 2. I still think it is important to give a deep analysis of the experiment results. For example, what is the reason that the model can obtain more robustness from the 'cat' class? Is it just because images in 'cat' classes are harder to learn? And why model obtain robustness for cars from cat images? They are very important questions to figure out the reason why we do not need to perturb all classes.
>
> 3. There is no theoretical analysis is a main flaw. For CIFAR-10, it is okay to study images in each class. And find the most efficient strategy. However, if there is no theoretical guarantee, it is difficult to design strategies for larger datasets, like ImageNet-1k. I hope the authors can prove they can efficiently find proper methods for ImageNet even without theoretical analysis.
>
> 4. I think that the authors misunderstand the question of generalizability. I mean that as the perturbed data are selected with the metrics entropy during the training process, it is possible that the perturbed data for the next training epoch will not be chosen. Because the model has the generalizability to give a clean version of the AE. Therefore, during the whole training process, all data could be selected to be perturbed, but they are selected in different epochs. If true, all data are perturbed, instead of only part of the training data.
>
> [1] [Adversarial training for free!](https://arxiv.org/pdf/1904.12843.pdf)

---

> > ### Author Response · Authors · 2023-11-20
> >
> > We thank the reviewer for their response and raising relevant additional questions.
> >
> > 1. **comparison with fast AT methods:** We kindly refer the reviewer to section A.9 in the appendix and its corresponding figure 22, where we discuss the combination of S-ESAT with single step Fast-AT on (Wong et al., 2020). The latter has been shown to be a more effective way of robustly training models (Wong et al., 2020).
> > We conclude that it is possible to further reduce training time without impairing robust accuracy.
> >
> >     Irrespective of the encouraging practical benefits, we want to reiterate again, that our paper is an *empirical analysis* of a phenomenon when adversarial training on a subset only (also see our answer to question 4). We see the improvements in training time as encouraging practical benefits, but do not claim them to be our main contribution.
> >
> >     Eric Wong, Leslie Rice, and J Zico Kolter. Fast is better than free: Revisiting adversarial training. ICLR, 2020.
> >
> >     (Also already cited in the main paper)
> >
> > 2. **missing deep analysis:** As stated before, we agree that such an analysis would be very valuable and we believe that such a deep dive will be imminent future work (perhaps conducted by the community). In that light, we would like to ask the reviewer whether they think that our paper has an interesting (potentially surprising) contribution that could spur follow up work. If yes, we would like to ask the reviewer to reconsider their recommendation.
> >
> > 3. **no theoretical analysis:** We politely object to the claim "*no theoretical analysis is a main flaw*". Some of the most important work in deep learning has no theoretical analysis on the why or how, but still remain important due to their empirical results -- concrete examples are the original AlexNet paper, or the more recent CLIP paper (to be clear, we do not want to claim that our submission is of comparable impact).
> >
> >     Furthermore, the questions raised by the reviewer: "For example, what is the reason that the model can obtain more robustness from the 'cat' class? [...] And why model obtain robustness for cars from cat images?" are indeed important, but the fact that our work raises these questions to begin with should count in our favour.
> >
> > 4. **entropy calculation:** We believe there is a misunderstanding regarding when we calculate the entropies and how we select $A$. As stated in the second sentence of section 3.3, "*prior to training, the training set is split into $A$ and $B$*" and this split remains fixed throughout training. Entropies are calculated *before* subset adv. training with a non-robust classifier, as stated in section 3.3, third paragraph and consequently does not change $A$ during training. Hence, SAT really only considers a fixed subset, such that *not* all data is perturbed.

---

> > > ### Comment · Reviewer_uHfk · 2023-11-21
> > >
> > > Thanks for the responses. I am sorry for the misunderstanding of how to divide the training set.
> > >
> > > 1. If I correctly understand the paper, the division requires a pre-trained model to calculate the entropy. Am I right? Is there any requirement for the pre-trained model? Will it be important to use the same model structure or optimizer?
> > >
> > > 2. It seems that Section A.9 and Fig. 22 are about transferring to other datasets. Could you provide the results of the same datasets?
> > >
> > > 3. Considering that there are some previous works discussing the adversarial properties of data in the dataset, such as [1], [2], [3], [4], [5], and [6], the finding that some data in the training set help the model improve the robustness and others do not is not very surprising. Therefore, I hope the authors could provide more analysis.
> > >
> > > 4. Thanks for the examples of AlexNet and CLIP. They are very excellent works. My concern is about the practical usage of this work. Because without a theoretical guarantee, we have no idea how it will work on a real-world large dataset. And AlexNet and CLIP achieve very impressive results on large datasets. Therefore, could the authors provide results on a large dataset, like ImageNet-1k, to address my concern?
> > >
> > > [1] Huang L, Zhang C, Zhang H. Self-adaptive training: beyond empirical risk minimization[J]. Advances in neural information processing systems, 2020, 33: 19365-19376.
> > > [2] Wang Y, Zou D, Yi J, et al. Improving adversarial robustness requires revisiting misclassified examples[C]//International conference on learning representations. 2019.
> > > [3] Zhang J, Xu X, Han B, et al. Attacks which do not kill training make adversarial learning stronger[C]//International conference on machine learning. PMLR, 2020: 11278-11287.
> > > [4] Zhang J, Zhu J, Niu G, et al. Geometry-aware instance-reweighted adversarial training[J]. arXiv preprint arXiv:2010.01736, 2020.
> > > [5] Dong Y, Xu K, Yang X, et al. Exploring memorization in adversarial training[J]. arXiv preprint arXiv:2106.01606, 2021.
> > > [6] Ge Y, Li Y, Han K, et al. Advancing Example Exploitation Can Alleviate Critical Challenges in Adversarial Training[C]//Proceedings of the IEEE/CVF International Conference on Computer Vision. 2023: 145-154.

---

> > > > ### Author Response · Authors · 2023-11-21
> > > >
> > > > We thank the reviewer for actively contributing to the discussion and appreciate the additional questions. However, we do not follow the reasoning regarding *theoretical guarantees*, which is something that is non-trivial in deep learning for complex vision tasks and hardly exists in literature. The reviewer for example refers to our interesting finding that robustness learned from cats transfers to cars but less so vice versa. Cats have unique visual characteristics vis-a-vis cars, e.g. (a) their highly non-rigid and articulated shape, (b) their non-uniform texture, and (c) the diverse viewpoints from which cat pictures are taken. It is not out of the question that these may play a role in the phenomenon we document in our work, but such visual attributes are highly non-trivial to formalize, especially with commonly used theoretical tools in (adversarial) machine learning. The latter typically treat images or image/object representations as highly abstract data points and make very loose assumptions about their content. While this suggests a need for novel theoretical tools that apply at a more appropriate level of abstraction, we feel that the challenges involved are under-appreciated and can be tackled by follow-up work -  unless you can point us to relevant work.
> > > >
> > > > We address each question below.

---

> > > > ### Author Response · Authors · 2023-11-21
> > > >
> > > > 1. Q2 - **fast AT:** We provide the requested results below for CIFAR-10 and CIFAR-100. We commit to providing results for ImageNet-200 in the final version. In our submission, we provided results for the task transfer setting, as this concretely shows that robustness is transferrable under single step AT even during task transfer.
> > > >
> > > >     Results with fast-ESAT on CIFAR-10 and CIFAR-100 show highly similar results to vanilla ESAT with $7$ PGD steps, except when $|A|\leq 10k$ for which robust accuray is near $0$%. This is likely due to catastropic overfitting (Wong et al., 2020, Andriushchenko et al., 2020), which could be mitigated with additional regularization (Andriushchenko et al., 2020). Note that we encountered this already for fast-S-ESAT in chapter A.9.
> > > >
> > > >     Andriushchenko, M. and Flammarion, N. (2020). Understanding and improving fast adversarial training. NeurIPS.
> > > >
> > > >     CIFAR-10:
> > > >
> > > >     |Method||5k|10k|15k|20k|25k|30k|35k|40k|45k|full AT|
> > > >     |--|--|--|--|--|--|--|--|--|--|--|--|
> > > >     |PGD-7|Rob. acc|52.1|59.1|63.3|64.9|66.4|66.8|67.7|68.7|68.5|69.0|
> > > >     ||Clean acc|91.9|91.2|90.5|89.7|89.4|89.4|89.2|89.4|89.5|89.2|
> > > >     |FGSM-RS|Rob. acc|0.0|51.1|59.8|62.9|64.1|64.4|65.0|66.6|65.8|--|
> > > >     ||Clean acc|75.7|91.2|91.0|90.2|90.3|90.2|89.4|89.8|90.0|--|
> > > >
> > > >     CIFAR-100:
> > > >
> > > >     |Method||5k|10k|15k|20k|25k|30k|35k|40k|45k|full AT|
> > > >     |--|--|--|--|--|--|--|--|--|--|--|--|
> > > >     |PGD-7|Rob. acc|21.0|28.8|33.4|36.3|38.0|38.7|39.5|40.0|40.7|40.9|
> > > >     ||Clean acc|73.8|71.8|70.4|68.5|67.8|67.0|66.1|65.8|65.8|65.9|
> > > >     |FGSM-RS|Rob. acc|0.2|2.1|28.5|33.0|35.4|36.9|37.6|38.8|39.2|--|
> > > >     ||Clean acc|77.5|76.8|71.1|69.5|68.4|67.8|66.9|66.6|67.2|--|
> > > >
> > > >
> > > > 2. Q1 - **stability of entropy estimates:** Very good question! To provide some insight into what requirements are needed for the model, we've evaluated fast-ESAT (single step ESAT) on CIFAR-100 with two different entropy estimates: (i) with estimates from a larger model, in this case ResNet-50 which achieves $77.8$% clean accuracy and (ii) with estimates from a much smaller model, in this case a 1-hidden layer multilayer perceptron (MLP) with 512 hidden linear units followed by ReLU. The latter achieving only $28$% clean accuracy.
> > > >
> > > >     We compare robust and clean accuracy of ESAT trained models in the table below and observe very similar characteristics between our vanilla experiment (using ResNet-18) and ResNet-50, both achieving above $35$% robust accuracy at $|A|=25k$. In contrast though, the small MLP model performs worse, achieving only $33.3$% robust accuracy at $|A|=25k$. While this experiment gives only a first insight, we recommend using a model of similar performance. For future work, it might be worthwhile to investigate whether we can use surrogate models models to cheaply estimate entropies on new datasets.
> > > >
> > > >     |Entropy source||5k|10k|15k|20k|25k|30k|35k|40k|45k|full AT|
> > > >     |--|--|--|--|--|--|--|--|--|--|--|--|
> > > >     |Vanilla (ResNet-18)|Rob. acc|0.2|2.1|28.5|33.0|35.4|36.9|37.6|38.8|39.2|40.9|
> > > >     ||Clean acc|77.5|76.8|71.1|69.5|68.4|67.8|66.9|66.6|67.2|65.9|
> > > >     |ResNet-50|Rob. acc|0.7|7.6|29.2|32.7|35.1|37.1|37.5|38.3|40.1|--|
> > > >     ||Clean acc|77.8|75.2|71.2|69.8|68.3|67.6|67.0|66.4|66.4|--|
> > > >     |MLP 512-100|Rob. acc|1.2|16.8|24.3|30.3|33.3|34.9|37.9|39.4|39.3|--|
> > > >     ||Clean acc|75.6|74.2|71.2|70.0|69.1|69.0|68.6|68.1|67.5|--|
> > > >
> > > > 3. Q3 - **results not surprising:** Thank you for providing references. While [2,3] or (Hua et al., 2021) do indeed show a link between example hardness and robustness gains (we briefly touched upon this in section 2 and section 3.3), none of them provide the insight that robustness transfers *that* well between classes and that this transfer can be counter-intuitive, depending on the class that is adv. trained. Overall, none of them have investigated AT under our subset setting such that a fixed data subset is never adv. perturbed. Furthermore, while the reviewer is adamant about the theoretical argument: we are not aware that any of the cited work provides theoretical *guarantees* about their claim. Why is the reviewer demanding this from our work?
> > > >
> > > > 4. Q4 - **ImageNet-1k:** We kindly refer the reviewer to our ImageNet-200 experiments, which are of similar scale and complexity to the ImageNet-1k dataset (Hendrycks et al., 2021). The demand to additionally scale to ImageNet-1k appears as arbitrary acceptance-threshold. We have shown that our results generalize from CIFAR-100 to ImageNet-200.

---

> > > > > ### Comment · Reviewer_uHfk · 2023-11-22
> > > > >
> > > > > Thanks for the updates and additional results.
> > > > >
> > > > > 1. Previous works [1,2,3,4,5,6] discuss data in the training set to help the model learn clean accuracy or robust accuracy. Therefore, I think they are related to this paper. It will be better to build a closer connection with these works. Especially, discussing the similarity of the data split in CSAT and ESAT between previous methods during the training process will be helpful.
> > > > >
> > > > > 2. ImageNet-1k contains more labels with similar semantic information. That is the reason that I would like to know how data in these classes will affect CSAT.
> > > > >
> > > > > 3. Based on the additional results, I would like to give a 6 for this paper and discuss it with AC and other reviewers. Nonetheless, I believe this will be an 8, if the authors could provide more analysis and theoretical guarantees.

---

> > > > > > ### Author Response · Authors · 2023-11-22
> > > > > >
> > > > > > We thank the reviewer for raising their score and for continuing to contribute to the discussion. We would like to provide the following additional discussion on the referenced related work for clarification and to comment on how we will integrate missing references into our manuscript.
> > > > > > We briefly state similarity/differences to SAT for each reference given:
> > > > > >
> > > > > > 1) *Huang L, Zhang C, Zhang H. Self-adaptive training: beyond empirical risk minimization. NeurIPS, 2020*:
> > > > > >
> > > > > >     Improves adv. training under data corruption. Perturbs *all* examples. No investigation on data subsets or impact of example hardness on robustness transfer.
> > > > > >
> > > > > > 2) *Wang Y, Zou D, Yi J, et al. Improving adversarial robustness requires revisiting misclassified examples. ICLR, 2019* (already cited in our paper):
> > > > > >
> > > > > >     Treats correctly and incorrectly classified examples differently during adv. training to improve robust accuracy. Perturbs *all* examples. Indirectly linked to our discussed example hardness, yet its ramifications for robustness transfer is not explored.
> > > > > >
> > > > > > 3) *Zhang J, Xu X, Han B, et al. Attacks which do not kill training make adversarial learning stronger. ICML, 2020*:
> > > > > >
> > > > > >     Propose early-stopped PGD, to increase adv. example hardness with training progression. Perturbs *all* examples. No direct link to clean example hardness though and no investigation of robustness transfer.
> > > > > >
> > > > > > 4) *Zhang J, Zhu J, Niu G, et al. Geometry-aware instance-reweighted adversarial training. ICLR, 2021:*
> > > > > >
> > > > > >     Loss weighting depending on distance to decision boundary, similar to our cited (Kim et al., 2021). Perturbs *all* examples. Very relevant paper that we will incorporate into our final manuscript. Does not discuss robustness transfer among classes and tasks. Only shows results on CIFAR-10 and SVHN.
> > > > > >
> > > > > > 5) *Dong Y, Xu K, Yang X, et al. Exploring memorization in adversarial training. ICLR, 2022*:
> > > > > >
> > > > > >     Shows models can memorize adv. examples, resulting in robust overfitting, by (partially) corrupting labels. Perturbs *all* examples. Overall no direct similarity to our SAT analysis, yet its findings could be potentially helpful in improving robust accuracies when $|A|$ is small, where overfitting is most likely to occur. We will incorporate this into our final manuscript.
> > > > > >
> > > > > > 6) *Ge Y, Li Y, Han K, et al. Advancing Example Exploitation Can Alleviate Critical Challenges in Adversarial Training. ICCV, 2023*:
> > > > > >     Similar to [2,4], propose to weight examples differently during adv. training depending on their contribution to clean or robust accuracy. Perturbs *all* examples. Does not discuss robustness transfer among classes and tasks. Evaluated only on CIFAR-10/100 and Tiny-ImageNet.

---

### Official Review · Reviewer_3h75 · 2023-10-31

**Soundness:** 3 good
**Presentation:** 3 good
**Contribution:** 3 good
**Rating:** 6
**Confidence:** 3

**Summary:**

To investigate the transferability of adversarial robustness across different classes and tasks, the authors proposed Subset Adversarial Training (SAT), which splits the training data into A and B and constructs adversarial examples (AEs) on A only.  Using SAT, this paper shows that training on AEs of just one class (e.g., cars in CIFAR-10) can transfer a certain level of robustness to other classes. Hard-to-classify classes (like cats) tend to provide greater robustness transfer compared to easier ones. Moreover, using AEs generated from half of the training data can match the performance of full AT. These findings also apply to downstream tasks. This paper distinguishes itself from others by only creating AEs on a pre-defined subset of the training set, independent of the model's architecture or the specifics of the training process.

**Strengths:**

1. The paper provides valuable insights into the transfer properties of adversarial robustness. The observation that adversarial robustness can transfer to classes that have never been adversarially attacked during training is intriguing.
2. The finding that generating AEs on merely 50% of the data can recover most of the baseline AT performance, especially on large datasets like ImageNet is insightful. This could potentially lead to significant computational savings without compromising the robustness of the model.
3. The paper's findings are not limited to a single task or dataset. The authors have undertaken a thorough experimental evaluation across multiple datasets.

**Weaknesses:**

1. While this paper presents intriguing empirical results on the SAT approach, it falls short of providing a clear explanation for the observed transferability of adversarial robustness from subset A to subset B.

**Questions:**

1. Can you provide more theoretical justification or intuitive explanations for the observed efficacy of constructing AEs on only a subset of the training data? Specifically, what underpins the phenomenon where harder examples offer better robustness transfer?
2. You mentioned that as dataset complexity increases, the trend of harder examples providing better robustness transfer diminishes. Can you explain the reasons behind this observation? Are there specific characteristics or properties of complex datasets that might be influencing this behavior?
3. Could you explain more on why the robustness transfer notably increases for smaller $\epsilon$ and decreases for larger $\epsilon$?

---

> ### Author Response · Authors · 2023-11-19
>
> 1. **no (theoretical) explanation for the observed transferability:**
> While we agree with the reviewer, that additional theoretical or empirical based explanations would be beneficial, we believe that our paper has a valuable contribution to the scientific community. We highlight a surprising phenomenon and counter-intuitive results -- e.g. (i) CIFAR-10 SAT on *cat* provides better robust transfer to *car* than *truck* on *car* or (ii) $30$% of training examples achieve baseline robust accuracies in the task transfer setting -- that could lead to new research directions.
>
>     1. **theoretical justification for why harder examples transfer robustness better:** At this point, we cannot provide an underlying theory. We speculate however, that classifying hard examples requires the detection of more diverse features, which are trained adv. robust. Given the compositional nature of deep networks, it is plausible to assume that these robust features are reused by other classes.
>
>     2. **reasons behind diminishing returns when dataset complexity increases:** We conjecture this to be similar to our point made for the previous question. Datasets with a larger number of classes and larger input images likely require the detection of more diverse features.
>
>     3. **why does robustness transfer increase for smaller $\epsilon$?:** We conjecture, that with increasing $\epsilon$ each feature becomes more class-specific and is thus less reusable. In contrast, with decreasing $\epsilon$ the generalizability is improved.

---

> > ### Comment · Reviewer_3h75 · 2023-11-20
> > **Official Comment by Reviewer 3h75**
> >
> > Thanks for the response of the authors. I have checked the updated version of the manuscript. After revision, the experiments become more solid. Besides, I agree with the authors that the major observations of this paper could lead to new research directions. In my opinion, the contributions of this paper are sufficient. Therefore, **I would recommend accepting this paper**. However, I am not fully convinced by the explanations provided by the authors as there is no theoretical guarantee. Overall, I will retain my current score.

---

> > > ### Author Response · Authors · 2023-11-22
> > >
> > > Dear reviewer 3h75,
> > >
> > > Thank you for the recommendation. In light of the recent discussion on theoretical guarantees with reviewer uHfk [see here](https://openreview.net/forum?id=pE6gWrASQm&noteId=y4WmyWsveU), we kindly ask the reviewer to reevaluate whether the requested theoretical *guarantees* are fair to request. We would appreciate a response.

---

### Official Review · Reviewer_gZFF · 2023-10-31

**Soundness:** 4 excellent
**Presentation:** 4 excellent
**Contribution:** 3 good
**Rating:** 8
**Confidence:** 4

**Summary:**

This paper conducts extensive experiments to see how only perturbing a subset of adversarial examples in training impacts adversarial robustness.  The authors find that using adversarial examples from certain classes can lead to nontrivial gains in robustness in other classes (despite not training with those classes).  The authors find that the most useful examples/classes to train with are correlated with their difficulty which they measure using entropy.

**Strengths:**

- paper is very clear
- great scope in experiments which encompass multiple datasets and model architectures
- experiments clearly demonstrate correlation between entropy and robust accuracy on the subset
- experiments suggest that in certain settings, we can train with a smaller subset of adversarial examples instead of all adversarial examples which can reduce the runtime of adversarial training, making it more feasible in practice.  Robustness also transfers to other datasets as well suggesting that this approach can be used with pretraining models.

**Weaknesses:**

- while it's clear that training with smaller subsets of adversarial examples can be beneficial, are there guidelines for how to determine the size of this subset to use SAT in practice?

**Questions:**

See weaknesses

---

> ### Author Response · Authors · 2023-11-19
>
> We thank the reviewer for the encouraging scores on our manuscript.
>
> 1. **guidelines for selecting subset size in practice?:**
>     We would like to emphasize that our proposed SAT methodology provides means to analyze the robustness transfer across classes and examples, yet for practical purposes we would give the following guidelines: for downstream task transfer, we recommend using the hardest $30$% of training data and otherwise $50$%. If calculating the entropies during training is intractable for whatever reason, it should be sufficient to use randomly sampled data.

---

### Official Review · Reviewer_QzqJ · 2023-11-01

**Soundness:** 3 good
**Presentation:** 3 good
**Contribution:** 2 fair
**Rating:** 6
**Confidence:** 3

**Summary:**

The paper studies the impact of adversarial training with a limited subset of data on the robust accuracy of computer vision models. The empirical study shows that a limited amount (30%) of carefully selected data is sufficient to achieve 90% of the robustness of the models. Additionally, the paper explores the transferability to other models of the robustness acquired with their method. Empirically, the paper shows that model robustness is best preserved with a custom-balanced loss and with hard-to-classify examples.

**Strengths:**

- The paper addresses the relevant problem of the generalization of the acquired robustness to unseen classes, examples, and tasks.
- The paper is well-written and easy to grasp.
- The experimental protocol is well described and allows to reproduce the study.
- The paper provides an in-depth empirical study to support its claim.
- The paper proposes a method to estimate the transferability of the acquired robustness which can be useful for testing ML models, as new classes and examples can appear after adversarial training in real-world systems.

**Weaknesses:**

I have a single but potentially critical concern regarding the significance of this work. In particular, I am unsure what are the benefits of considering a subset of the adversarial examples during adversarial training to improve the efficiency of the process.

As I understand, the key objective of this paper is to limit the size of the set of examples used for adversarial training, to achieve a similar robust accuracy than with full adversarial training. The reason is that full adversarial training is computationally expensive. Considering the settings described in the experimental protocol, I fail to understand why the proposed process is more efficient than full adversarial training. In both cases, the full datasets need to be labeled and the same amount of computational resources is needed, as only the set of available examples for adversarial training differs, not the total number of examples used during adversarial training (since the subset of adversarial examples is completed with the clean examples not used for generating adversarial examples. One could argue that we save 70% of the time to generate the adversarial examples, but the largest cost still come from model training. Considering the empirical results that demonstrate that the proposed method does not lead to better robust accuracy than full adversarial training, I do not see the added value for model robustness efficiency/effectiveness. It would be good if the authors could clarify the benefits of using only a subset of examples in the adversarial training process.

**Questions:**

- What is the exact process of adversarial training used in the experiments? Are all examples adversarially perturbed? Are some examples perturbed and others clean? In what proportion? In total is the number of perturbation executions the same for adversarial training as subset adversarial training?

- What is the objective of transferring the robustness across classes and tasks? We need to retrain models to adapt to these new classes and tasks, isn’t it simpler to adversarial train them at retraining, therefore empirically obtaining a more robust model ?

Other comments:

- On page 6,  lines 6 and 11, of the paragraph left to fig 3, describe the results for the class “car” while the numerical value corresponds to the line “plane” on fig 2.
- Figures are not readable when printed.

---

> ### Author Response · Authors · 2023-11-19
>
> 1. **what are the benefits of considering a subset:?**
>
>     The benefits of considering a subset are twofold: (a) practical, and (b) analytical.
>
>     **Practical:** We observe significant savings in terms of training cost with our protocol. For example, training our ResNet-18 models on CIFAR-10 takes approx. 3h without any adversarial perturbations. Full adversarial training takes about 7h, and only perturbing 50\% of the data results in a training time of 5h. Thus  reducing the cost of training considerably. These savings are expected to be even larger at scale, that is when training large models on large datasets. E.g. consider the power consumption of training foundational models, which consume hundreds of Mega-Watt-hours without adversarial training (e.g. nearly 500MWh for LLaMa-65B [LLama]). Adding adversarial training roughly increases the training time -- and thus the power consumption -- by a factor of PGD-steps. Our insights suggest that training time/consumption could be at least halved.
>
>
>     **Analytical:**
>     While it is true that "the proposed method does not lead to better robust accuracy than full adversarial training", we show that you can trade-off a minor loss in robust accuracy against significant savings in training time -- by not perturbing all classes/examples. However, we also believe that this loss in robust accuracy should motivate further research into more effective defenses against adversarial attacks: Why **do** we need to perturb all classes/examples to achieve the full potential of adversarial training? Can we train models to be robust in a class-agnostic manner? These are important questions raised by our work, which we believe to be of interest to the scientific community.
>
>     [LLama] Touvron, Hugo, et al. "Llama: Open and efficient foundation language models." arXiv preprint arXiv:2302.13971 (2023).
>
> 2. **adversarial training protocol:** For vanilla adversarial training, we perturb all training examples. There is no mix between clean and perturbed examples in a batch as described in earlier work on adv. training. For SAT on the other hand, we randomly sample a training batch and perturb only those examples that are contained in $A$. We have updated the appendix, section A.1 in our manuscript to make this clear.
>
> 3. **during task transfer, why not retrain with full adv. training?** Our argument is similar to question 1. We see an increase in foundational models, which provide strong off-the-shelf performance which can be improved via fine-tuning. Given the computational demand that is required to train such large models, it is often sufficient to only train a shallow classifier on top of intermediate outputs.
> In this case, adversarial robustness can only be provided when the foundational model is robust.
> We have shown that it is sufficient to perturb only $30\%$ of examples during the initial (foundational) training to achieve high adversarial robustness after fine-tuning.
>
> 4. **mismatched reporting on page 6:** Thank you for highlighting this issue. We have fixed it in the new revision. Note that, now, the numerical values are actually better than what was reported in our initial text.
>
> 5. **unreadable figures:** Thank you for highlighting a readability issue. We will improve the readability for the final revision.

---

> > ### Comment · Reviewer_QzqJ · 2023-11-23
> >
> > Dear authors,
> >
> > Thank you for the detailed answer. The full and partial AT protocols are now clearer. With the described protocol, the empirical and theoretical description of the efficiency gain of the approach makes sense. Subset AT has the potential to drastically reduce adversarial training costs (both in classic and transfer learning settings).
> >
> > I share the concerns of reviewers gZFF and uHfk on the practical application of this method on larger, unknown datasets, while maintaining the time benefits. Indeed we do not know a priori the size of the subset on which to apply AT. I am not convinced that the simple guidelines (30% and 50%) given as an answer to reviewer gZFF will generalize.
> >
> > Additionally, from the answer given to reviewer uHfk (20.11.2323 14:13) on fast AT methods, there are - potentially more effective - methods that have been proposed for the same problem but the paper does not compare with them.
> >
> > I will continue following the discussion on these matters.

---

> ### Author Response · Authors · 2023-11-23
>
> Dear reviewer, thank you for the response.
>
> We would like to emphasize that our paper should not **exclusively** be read as providing “guidelines“ for more efficient adversarial training. As described earlier in this thread, there is also an analytical component to our paper centered on the question: Do we need to adversarially perturb images from every class to attain good robustness, and what are the implications?
>
> There is a “glass half full” answer that we back up with our experiments, namely that one can get away with a minor loss of robustness by **not** perturbing examples from every class. This has positive practical implications that we have discussed at length.
>
> There is however also an interesting “glass half empty” answer, namely that to achieve the full potential of adversarial training in terms of robustness, it would seem that one **indeed** has to perturb representative images from every class, but also that the choice of classes is inferior when random. This has all kinds of interesting implications and suggests further questions and directions for research, e.g. to name a few: While we provide some measures that correlate with robustness transfer, what really are the underlying mechanisms? What theoretical tools are needed to distinguish between individual object and scene classes in terms of their visual characteristics? What does this say about adversarial training as a general framework if robustness ultimately has to be learned for every class? Can we be satisfied with visual recognition methods that must gain robustness in this manner? Is this scalable?
>
> Note: Regarding the fast adversarial methods (esp Wong et al 2020) our initial experiments show that these may well be complementary but we acknowledge that further investigation is warranted.

---

### Author Response · Authors · 2023-11-19
**Comment to all reviewers**

We thank all reviewers for their time and valuable feedback on our manuscript. We are very pleased to read that our paper "*could potentially lead to significant computational savings without compromising the robustness of the model*" (3h75), that our scope in experiments is great and in-depth (QzqJ,gZFF,3h75), that soundness was rated excellent (RgZFF), that we address a relevant problem and provide valuable insights (QzqJ,gZFF,3h75,uHfk) and that our paper is very clear and well written (QzqJ,gZFF,uHfk).
We address each reviewer in individual comments and have updated our manuscript. Each change is marked in red.

---

### Comment · Area_Chair_MZKx · 2023-11-21
**[Time Sensitive, ICLR24] Please read the authors' responses and try to discuss the remaining concerns with the authors**

Dear Reviewers,

The authors have provided detailed responses to your comments.

Could you have a look and try to discuss the remaining concerns with the authors? The reviewer-author discussion will end in two days.

We do hope the reviewer-author discussion can be effective in clarifying unnecessary misunderstandings between reviewers and the authors.

Best regards,

Your AC

---

### Author Response · Authors · 2023-11-23
**Final remarks**

Dear Reviewers, dear AC,

Thank you for providing constructive feedback on our manuscript. We firmly believe this has made our submission stronger and would like to extend our sincere gratitude to all of you for the engaged discussion and willingness to consider our responses.

Approaching the end of our discussion phase, we would like to take the opportunity to briefly summarize the main insights of this rebuttal:

1. Our work considered subset adv. training (SAT), perturbing only a fixed training subset $A$. This consideration has two benefits (a) practical and (b) analytical.

    **Practical:** Substantially reduce adv. training time, even under single step AT (FGSM, as described in Wong et al., 2020)

    **Analytical:** SAT provides a framework to analyze whether/why we need to perturb all classes/examples to achieve the full potential of AT.

2. TRADES can complement SAT to trade-off clean accuracy (which increases when $|A|$ is small) for increased robust accuracy. Only considering robust accuracy, SAT+TRADES reaches baseline AT performance when $|A|$ contains $40$% of training data.

3. Discussion on (theoretical) analysis: While we agree with reviewers that additional analysis is needed on why particular classes provide better robustness transfer than others, we've argued that it is implausible to give explicit theoretical guarantees that our method generalizes to larger datasets. Irrespectively, we want to emphasize, that we've shown generalization to ImageNet-200 and that it is possible to join such a complex problem with fast (single step) adv. training.

---

> ### Comment · Area_Chair_MZKx · 2023-12-02
>
> Dear authors,
>
> Your comments will be fully considered in the final recommendation. Many thanks for your efforts made during the rebuttal.
>
> Best regards,
>
> Your AC

---

### Meta-Review · Area_Chair_MZKx · 2023-12-05

**Metareview:**

This paper makes an interesting contribution by analyzing how adversarial robustness transfers across subsets and classes. The obtained observation results are insightful, which might motivate more work in the AT field. However, mathematical analysis is not solid and should be strengthened. Please note that experimental verification is not equivalent to theoretical proofs.

**Justification For Why Not Higher Score:**

More theoretical analysis should be added to this paper. Experiments are solid but cannot replace with theoretical results.

**Justification For Why Not Lower Score:**

This paper indeed contributes interesting observations to the field, which might lead to more follow-up ideas. Thus, this paper can be accepted by ICLR.

---

### Decision · Program_Chairs · 2024-01-16

Accept (poster)